# Nonstoichiometric acid–base reaction as reliable synthetic route to highly stable $CH_3NH_3PbI_3$ perovskite film

Mingzhu Long[1,*], Tiankai Zhang[1,*], Yang Chai[2], Chun-Fai Ng[3], Thomas C.W. Mak[3], Jianbin Xu[1] & Keyou Yan[1]

Perovskite solar cells have received worldwide interests due to swiftly improved efficiency but the poor stability of the perovskite component hampers the device fabrication under normal condition. Herein, we develop a reliable nonstoichiometric acid–base reaction route to stable perovskite films by intermediate chemistry and technology. Perovskite thin-film prepared by nonstoichiometric acid–base reaction route is stable for two months with negligible $PbI_2$-impurity under $\sim 65\%$ humidity, whereas other perovskites prepared by traditional methods degrade distinctly after 2 weeks. Route optimization involves the reaction of $PbI_2$ with excess HI to generate $HPbI_3$, which subsequently undergoes reaction with excess $CH_3NH_2$ to deliver $CH_3NH_3PbI_3$ thin films. High quality of intermediate $HPbI_3$ and $CH_3NH_2$ abundance are two important factors to stable $CH_3NH_3PbI_3$ perovskite. Excess volatile acid/base not only affords full conversion in nonstoichiometric acid–base reaction route but also permits its facile removal for stoichiometric purification, resulting in average efficiency of 16.1% in forward/reverse scans.

[1] Department of Electronic Engineering, The Chinese University of Hong Kong, Shatin, New Territories, 999077 Hong Kong, China. [2] Department of Applied Physics, The Hong Kong Polytechnic University, Hung Hom, 999077 Hong Kong, China. [3] Department of Chemistry, The Chinese University of Hong Kong, Shatin, New Territories, 999077 Hong Kong, China. * These authors contributed equally to this work. Correspondence and requests for materials should be addressed to J.X. (email: jbxu@ee.cuhk.edu.hk) or to K.Y. (email: kyyan@ee.cuhk.edu.hk).

Solution processed perovskite solar cells (PSC) using $AMX_3$ ($A = Cs^+$, $CH_3NH_3^+$ ($MA^+$) or $NH = HCNH_3^+$ ($FA^+$); $M = Sn^{2+}$, $Pb^{2+}$, $Ge^{2+}$; $X = Cl^-$, $Br^-$, $I^-$) as light absorber have received broad interest in photovoltaics and light-emitting application in the past 5 years due to their ease of fabrication, cost-effectiveness and high efficiency. In terms of photovoltaic performance, it exceeds 22% power conversion efficiency in a short time, which is competitive with long-term developed thin-film-based solar cells (such as CIGS, polycrystalline Si solar cell)[1–10]. As confirmed, perovskite has a 100–1,000 nm diffusion length in polycrystalline films and over 100 microns diffusion length in single crystals, affording 100% internal quantum efficiency in thin-film-based PSC[11–14]. In combination with much higher photovoltage and light extinction coefficient, PSC can potentially outperform the efficiency of silicon solar cell in the foreseeable future[11,14–16].

However, as organic–inorganic ionic crystals with intrinsic humidity and thermal instabilities, perovskites encounter difficulties in large-scale application and commercialization. Recently, a layered and/or two-dimensional perovskite as well as pseudo-halide perovskite light absorbers were demonstrated to possess enhanced moisture stability, but unfortunately the efficiency was low (2–8%), probably due to poor electronic properties attributable to their long-chain cation and large-size anion components[17–19]. Besides, a crystal crosslinking strategy was performed with improved stability at ~55% humidity in the dark, retaining 9–10% *PCE* after 1,000 h under 10% one-sun illumination[20]. Moreover, Chen *et al.*[21] developed a stable PSC adopting highly doped inorganic charge extraction layer with shielding capacity from moisture and delivered 15% *PCE* with 1 cm² large area. However, the stability of perovskite itself without shielding was still <1 week in a humid environment[20,21]. Without device encapsulation, MAPbI₃/FAPbI₃ perovskites were regarded to be unstable under high humidity. The degradation kinetics had been studied to some extent using thermal gravity analysis (TGA)[22] and ultrafast spectroscopy[23], which revealed some possible routes, but the degradation process related to transition states and intermediate products is not well-resolved. For example, the sequence of HI and $CH_3NH_2$ release was ambiguous and some believed that HI was released before $CH_3NH_2$ (ref. 24). Besides, $CH_3NH_3I$ was very thermally stable even when the temperature increased to 150 °C (refs 22,25), but the perovskites quickly decayed to $PbI_2$ in association with $CH_3NH_2$/HI release at 80–150 °C after 24 h, which meant the decomposition kinetic pathways were different from each other[26]. Thus, this degradation kinetics of perovskite requires a closer examination to address these stability issues, and an alternative strategy is expected for enhancement of perovskite stability while keeping its high performance.

In this work, we report our investigation of the degradation and recovery of $CH_3NH_3PbI_3$ perovskite, which revealed the role of the intermediate product as well as methylamine amount in achieving stability. Hybrid perovskite decomposes sequentially in terms of thermodynamics based on MA-recoverable degradation, such that it first loses $CH_3NH_2$ and then HI, leaving behind $PbI_2$ solid. On the basis of this, we have developed an alternative two-step nonstoichiometric acid–base reaction route (NABR) for the synthesis of moisture-resistive perovskite, that is the production of starting $HPbI_3$ using excess HI to react with $PbI_2$, followed by perovskite conversion from $HPbI_3$ using excess $CH_3NH_2$. We have established that $CH_3NH_2$ abundance in synthesis and the formation of high-quality $HPbI_3$ built of columnar face-sharing $PbI_6$ octahedra are two key parameters for stabilized perovskite. In NABR, excess and volatile reagents lead to complete reaction as well as stoichiometry, respectively. Therefore, a perovskite thin-film prepared via this route possesses

reduced lattice vacancy, thus eliminating the penetration of undesirable $H_2O$ molecules into vacant sites and avoiding formation of monohydrate degradation product in association with H-bonding between $H_2O$ and $MA^+$. We have demonstrated that the $CH_3NH_3PbI_3$ perovskite thin-film remained highly stable in ~65% humidity for up to 2 months with negligible $PbI_2$-impurity, whereas other perovskites prepared by traditional one-step or two-step methods degrade distinctly after 2 weeks. The PSC using thin-film after humidity exposure delivered even better efficiency than that from freshly prepared film probably due to slight doping. This work provides an important insight into perovskite intrinsic stability and the utilization of simple chemical reaction for material control in PSC.

## Results

**Recoverable degradation.** To find a way to improve the stability of $CH_3NH_3PbI_3$, we have first investigated the degradation of traditional perovskites and used the recent defect-healing process for the recovery of degraded perovskite to check the transition products[27]. Figure 1a–c show the X-ray diffraction, optical images and photoluminescence mapping for degraded and recovered perovskite thin-film (prepared by traditional two-step method). We found that after degradation for some time, perovskite became yellow (Fig. 1b left) and had the distinct X-ray diffraction pattern of $PbI_2$ at $2\theta = 12.6°$ (Fig. 1a). Although we thought it was fully degraded, it was immediately recovered to great extent by exposure to $CH_3NH_2$ vapour (Fig. 1b right). The $CH_3NH_2$-recovered perovskite film was confirmed by the X-ray diffraction peaks (Fig. 1a) at $2\theta = 14.1°$, 28.4° that were indicative of $CH_3NH_3PbI_3$ (110), (220) facets. The recovered perovskite films existed in the form of nanoscale crystals judging from the broad X-ray diffraction peaks. The photoluminescence mapping contrasts (Fig. 1c) for the degraded (left) and recovered (right) films indicated their uniform reversion to perovskite. In comparison, the degraded film prepared by traditional one-step method was used for recovery test. These films had fast degradation rates and yielded some transition product under observation during the degradation. Figure 1d–f show the basic results. After 3-day degradation in 65% humidity, we observed a small-angle X-ray diffraction peak at $2\theta = 8.1°$, which was identical to that in the monohydrate ($CH_3NH_3PbI_3 \cdot H_2O$, with X-ray diffraction peaks at $2\theta = 8.10°$, 8.66° and 10.66°) (refs 23,25,28). After 3 weeks, the degraded film had only one $PbI_2$ peak at $2\theta = 12.6°$. However, these films were partially recovered as perovskite nanoscale crystals (Fig. 1d), with photoluminescence mapping so as to confirm the uniform recovery judged from the strong photoluminescence at 760 nm (ref. 29; Fig. 1f). The similar process of degradation and recovery was also observed in the mixed $CH_3NH_3PbI_{3-x}Cl_x$ perovskite using 3:1 mole combination of $CH_3NH_3I$ and $PbCl_2$, which was high-performing in PSC but presumably the most unstable perovskite compared with the iodide perovskite. We saw that the degradation process clearly exhibited the transition monohydrate product of $CH_3NH_3PbI_{3-x}Cl_x \cdot H_2O$ in the degradation, as shown in small angles of X-ray diffraction at $2\theta = 8.10°$, 8.66° and 10.66°, corresponding to the (001), (100) and $(10\bar{1})$ reflections of a monoclinic P2₁/m crystal structure, and could be recovered to some extent (Supplementary Fig. 1a). The monohydrate is similar to $CH_3NH_3PbI_3 \cdot DMF$ (dimethylformamide), in which the $MA^+$ is connected with solvent molecules ($H_2O$/DMF) through H-bonding (Supplementary Fig. 1b,c) and thus degradation reaction occurs.

To evaluate the reaction process of perovskite more clearly, large crystals were prepared by immersing $PbI_2$ into 2–5 mg ml$^{-1}$ MAI IPA solution for *in situ* observation of the recovery process

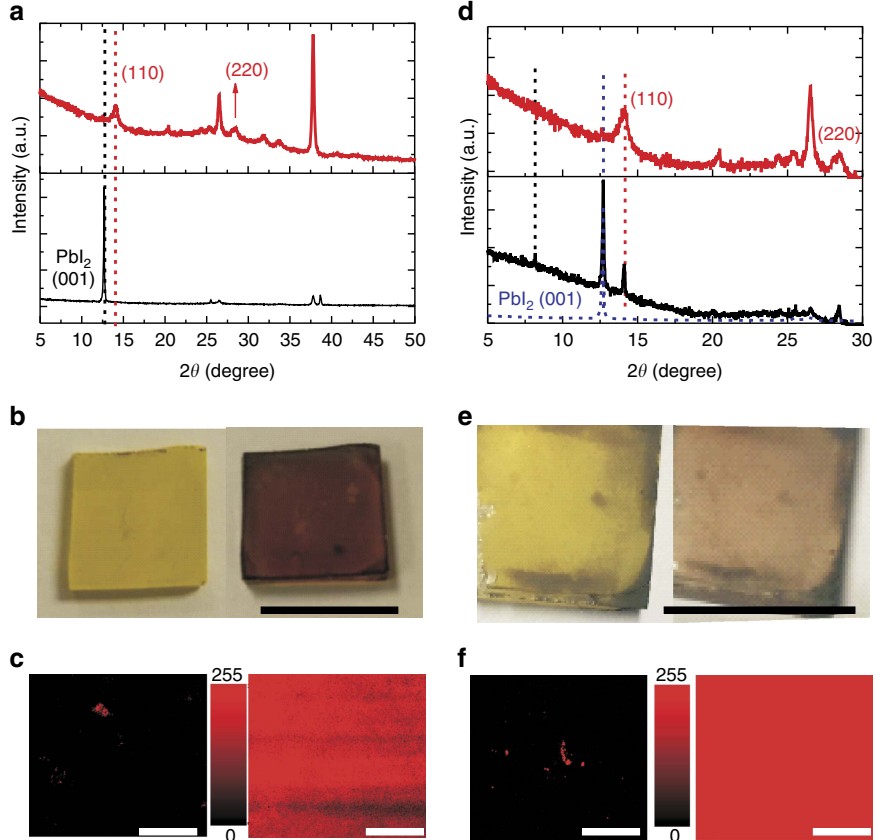

**Figure 1 | Recoverable perovskite degradation.** (**a**) X-ray diffraction patterns of degraded (black line) and recovered film (red line) prepared by traditional two-step method. (**b**) Photographs and (**c**) photoluminescence mapping of degraded perovskite film (left) and recovered perovskite film (right), for a thin-film prepared by two-step method, with the colour bar in the middle representing photoluminescence in counts. (**d**) X-ray diffraction patterns of degraded films under 3 days moisture (black line) and 21 days moisture (blue dash line) and recovered film (red line) for a thin-film prepared by traditional one-step method. (**e**) Photographs and (**f**) photoluminescence mapping of degraded perovskite film (left) and recovered perovskite film (right) prepared by one-step method, with the colour bar in the middle representing photoluminescence in counts. Scale bar, 1.5 cm (**b**,**e**), 50 μm (**c**,**f**).

via optical microscope. We found that after degradation, the yellow phase crystals did not differ much in morphology from that of the parent black perovskite (Supplementary Fig. 2). After recovery using $CH_3NH_2$, the degraded crystals re-crystallized as much smaller and more compact grains (Supplementary Fig. 3). This general recovery suggested that the degraded film with $PbI_2$ and monohydrate could be re-converted to perovskite using methylamine and thus inferred that degradation of perovskite was also related to the loss of methylamine.

We have analysed the chemical reactions for perovskite formation in DMF solution and degradation with the following scheme:

$$PbI_2 + CH_3NH_3I \xrightarrow{DMF} PbI_3^- \cdot xDMF + CH_3NH_3^+ \quad (1)$$

$$\leftrightarrow HPbI_3 \cdot xDMF + CH_3NH_2 \quad (2)$$

$$\xrightarrow{\text{spin coating}} CH_3NH_3PbI_3 \cdot DMF \quad (3)$$

$$\xrightarrow{\Delta} CH_3NH_3PbI_3 \quad (4)$$

On the basis of previous work[25,28], we observed the colloidal characteristics and redshift of perovskite precursor compared with $PbI_2$, which verified that either (1) or (2) was right, which could yield $CH_3NH_3PbI_3 \cdot DMF$(3). The intermediate $CH_3NH_3PbI_3 \cdot DMF$ (Supplementary Fig. 1c,d) has been detected by X-ray diffraction in another report[28] and will be confirmed in

the following. Noted that these routes could yield possible byproducts shown below due to coordination of DMF to Pb(II) in the solution accompanied by its subsequent removal in the film:

$CH_3NH_2$ vacancy $\qquad (CH_3NH_2)_x HPbI_3$

Iodide vacancy or both $\quad (CH_3NH_2)_x H_y PbI_{2+y} \quad 0 < x, y < 1$

In the degradation, we could simply consider the following step reactions:

$$CH_3NH_3PbI_3 + H_2O \xrightarrow{\text{moisture}} CH_3NH_3PbI_3 \cdot H_2O \quad (5)$$

$$\xrightarrow{\text{moisture}} HPbI_3 + CH_3NH_2 \cdot H_2O \uparrow \xrightarrow{\text{moisture}} HPbI_3 + H_2O + CH_3NH_2 \uparrow \quad (6)$$

$$\xrightarrow{\text{moisture}} PbI_2 + HI \uparrow + H_2O \uparrow \quad (7)$$

Actually, reaction (5) has been verified through *in situ* time-resolved X-ray diffraction techniques and heating-recovery in a previous study[30] and reaction (7) can be easily concluded from the final products. Degradation reaction (6) was apparently judged from $CH_3NH_2$-recovery in the glovebox (Fig. 1 and Supplementary Fig. 3) and directly proved in the following investigation. Therefore, we conclude that in the humidity degradation, the perovskite is sequentially decomposed in terms of thermodynamics, first forming an intermediate mono-hydrate (5), then liberating $CH_3NH_2$ molecules (6), and finally

yielding a $PbI_2$ solid and releasing $HI/H_2O$ vapour (7), although the kinetic pathways could produce different final products (for example, $CH_3NH_3PbI_3 \cdot H_2O$, $PbI_2$ and $HPbI_3$, and so on)[23,30]. Hence, we can make two conclusions. First, from the formation and degradation analysis, we can see that the formation of a fully coordinated perovskite $[PbI_3]^-$ framework is the first important parameter, which can reduce the I-vacancy. Second, as $CH_3NH_2$ can recover the degraded perovskite through the reverse reactions of (5–7), $CH_3NH_2$ abundance is able to improve the stability of perovskite through retarding the degradation reaction. Therefore, the key to stable perovskite is to build a fully coordinated and robust $[PbI_3]^-$ scaffold that accommodates $CH_3NH_2$ at A sites of perovskite $AMX_3$ lattice, thus eliminating the inclusion of undesirable water molecules and suppressing the degradation reaction.

**Acid–base reaction.** To synthesize stable perovskite, we propose a two-step route through NABR based on the above analysis where $\Delta_r G_1$ and $\Delta_r G_2$ are the changes of Gibbs free-energy for reaction (8) and (9), respectively.

$$PbI_2 + HI \xrightarrow{DMF} HPbI_3 + \Delta_r G_1 \qquad (8)$$

$$HPbI_3 + CH_3NH_2 \rightarrow CH_3NH_3PbI_3 + \Delta_r G_2 \qquad (9)$$

In reaction (8), excess hydriodic acid promotes the reaction completely and yields $HPbI_3$ that is stoichiometrically identical to the $[PbI_3]^-$ framework of perovskite, permitting it to form perovskite without I-vacancy. In reaction (9), excess $CH_3NH_2$ allows complete conversion of $HPbI_3$ to perovskite and thus eliminates the $CH_3NH_2$ vacancy. Different from the use of excess $CH_3NH_3I$ to react with $PbI_2$, both $CH_3NH_2$ and HI are facile for removal to yield vacancy-free perovskite due to their volatility at room temperature.

Faint yellow $HPbI_3$ crystal shows a hexagonal array of $[PbI_3]^-$ anionic columns each composing of $[PbI_6]$ coordination octahedra that are stacked along their opposite triangular facets, with exact molecular formula $HPbI_3$. Presumably, the protons in $HPbI_3$ can move freely in the intervening space between hexagonal arrays of anionic coordination columns. Powdery $HPbI_3$ as starting material for device fabrication was prepared following the literature procedure[31], with careful modification using ethanol instead of diethyl ether to remove excess HI and intercalated solvents ($H_2O$, DMF) and precipitate our products. (see the Methods section and Supplementary Fig. 4).

All the lead polyiodide precursors in DMF exhibited colloidal behaviour judging from their Tyndall effects using 532 nm laser beam (Fig. 2a). $PbI_2$ was coordinated by DMF ligands for dissolution and formed colloids in high concentration[25]. Precursors of both $HPbI_3$ and 1:1 combination of $CH_3NH_3I:PbI_2$ had smaller colloidal size judging from weaker light scattering than that of $PbI_2$ due to increased coordination number of Pb(II) (Fig. 2a). Therefore, $HPbI_3$ had the advantage of solubility in much higher concentration ($>2.5$ M) than $PbI_2$. We compared the absorption spectra of $HPbI_3$, $PbI_2$ and 1:1 $CH_3NH_3I:PbI_2$ (one-step precursor) and found that those of $HPbI_3$ and 1:1 $CH_3NH_3I:PbI_2$ had 20–30 nm red-shifted absorption edge compared with that of $PbI_2$, suggesting that coordination occurred between iodide and $PbI_2$ in both $HPbI_3$ and the 1:1 complex $CH_3NH_3I:PbI_2$ (Fig. 2b). Moreover, the absorption of $HPbI_3$ solution was slightly red-shifted with respect to that of 1:1 $CH_3NH_3I:PbI_2$ solution, indicating complete coordination due to the use of excess HI. In the NABR precursors, the addition of too much $CH_3NH_2$ ethanol solution decreased the solubility in the mixed solution and precipitated into some solid phase (Fig. 2c).The increased beam size and distinct light scattering effects were monitored, which were used for optimizing organic–inorganic ratios of $CH_3NH_2:HPbI_3$ (ref. 25). Due to acid–base neutralization (reaction (9)), $CH_3NH_2$ could be spontaneously arranged in the $HPbI_3$ framework. With increasing addition of $CH_3NH_2$ solution, the absorption began to exhibit a blue-shift (Fig. 2d). This blue-shift was actually the bleaching state of perovskite compound with excess $CH_3NH_2$ intercalation inside, which thus did confirm that the formation towards perovskite configuration started in the solution and could be readily converted to perovskite after DMF removal[24,32].

As control experiment, we also added $CH_3NH_2$ solution into a 1:1 molar solution of $CH_3NH_3I:PbI_2$. We observed not only the blue-shift suggesting $CH_3NH_2$ intercalation but also the strong absorption peak at 335 nm indicative of $PbI_2$ monomer, which meant it lacked iodine coordination in this recipe[33] (Fig. 2e). This iodine deficiency was consistent with a slightly bluer absorption edge than aforementioned $HPbI_3$ solution, which would lead to iodine vacancy by the control method.

**Crystalline phase conversion.** We first checked the morphology of thin films (Fig. 3). $PbI_2$ film prepared from DMF solution did not show the anisotropic rod-shape of its powders that should be essentially in accordance with the trigonal phase, but afforded uniform coverage and flatness on the substrates due to DMF coordination (Fig. 3a). $HPbI_3$ deposited as large-sized bundle-like crystals from solution by spin-coating, with poor coverage in film formation due to Ostwald ripening (Fig. 3b). In the control method, 1:1 combination of $CH_3NH_3I:PbI_2$ produced dendritic bundles of perovskite (Fig. 3c), which is related to $CH_3NH_3PbI_3 \cdot DMF$ solvate and will be discussed later. In NABR, perovskites precipitated in the similar morphology to that from the control method due to similar iodide coordination (Fig. 3d–f). However, $CH_3NH_2$ was able to form a liquid interface ($HPbI_3 \cdot x(CH_3NH_2)$) with perovskite[27] through intercalation/coordination and thus served as surfactants to refine and passivate the grain. Therefore, the increased $CH_3NH_2$ in NABR could reduce the crystal size (Fig. 3e,f) and improve film coverage.

We have correlated the morphology with crystallographic information in detail through X-ray diffraction monitoring the conversion (Fig. 4). Dip-coated wet film, spin-coated films before baking and after baking are deposited to represent the three basic conversion stages. The wet $PbI_2$ film had small-angle peak at $2\theta = 9.5°$, which is due to ligand behaviour of DMF ($PbI_2 \cdot DMF$; Fig. 4a black curve) that is similar to that of dimethyl sulfoxide (DMSO) in $PbI_2 \cdot DMSO$[9]. This soft coordination complex facilitated film formation of $PbI_2$ as aforementioned through gradual release of coordinated DMF during spin-coating without baking (Fig. 4a blue dashed curve). Baking increased the crystallinity from the strong (001) peak at $2\theta = 12.6°$ (Fig. 4a blue solid curve). $HPbI_3$ film displayed thin-film X-ray diffraction peaks at $2\theta = 11.5°$, $15.8°$, $20.1°$, $25.8°$, corresponding to (100), (101), (110) and (201) facets of hexagonal $HPbI_3$, respectively. Even when the film was a little wet, it displayed the same indexed X-ray diffraction pattern as the dry films without/with baking after DMF loss, which is consistent with the single-crystal X-ray diffraction analysis on their solvate and dry crystal (Fig. 4b and Supplementary Fig. 4). The transition products of perovskites were recognized by their small-angle X-ray diffraction peaks in wet films at $2\theta = 6.5°$, $7.9°$ and $9.4°$, corresponding to monoclinic $CH_3NH_3PbI_3 \cdot DMF$ (also see Supplementary Fig. 1c,d). However, in the spin-coated films before baking, these small-angle peaks of NABR film disappeared, suggesting its complete conversion to perovskite (Fig. 4c), but the control one-step sample still showed

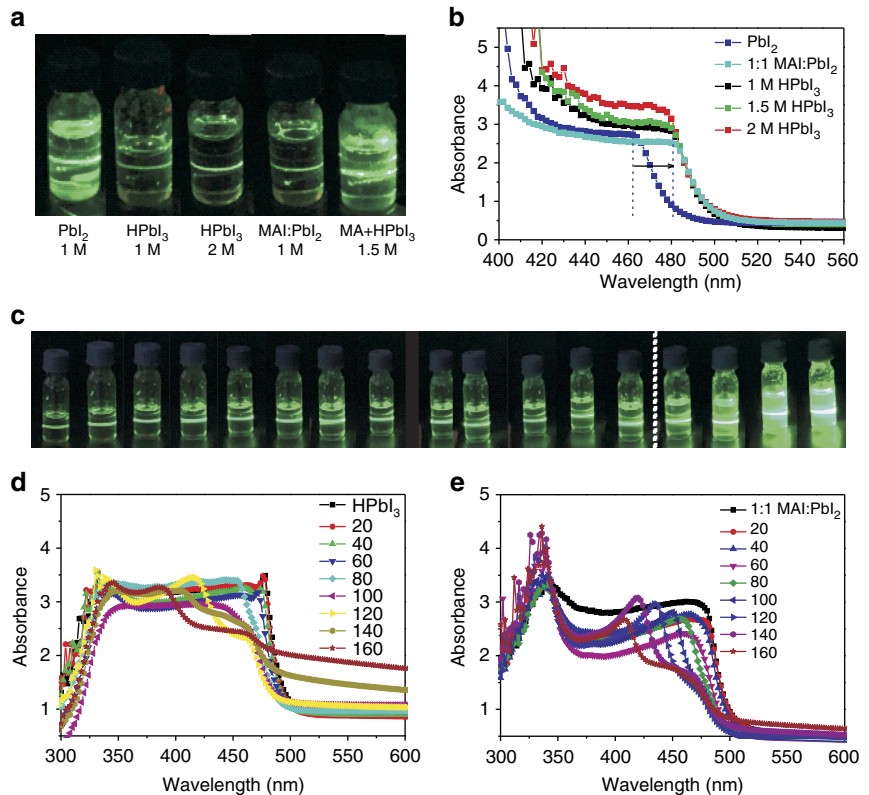

**Figure 2 | Acid–base reacted colloidal solution.** (**a**) Tyndall effect using 532 nm laser suggests the colloidal behaviour of starting 1.0 M $PbI_2$ and 1.0 M, 2.0 M $HPbI_3$, control 1:1 combination of 1 M $CH_3NH_3I:PbI_2$ precursor and acid–base precursors prepared via adding 0.2 ml $CH_3NH_2$ ethanol solution into 0.5 ml of 1.5 M $HPbI_3$ solution. (**b**) Ultraviolet spectra of starting $PbI_2$ and $HPbI_3$ precursors, as well as control 1:1 combination of 1 M $CH_3NH_3I:PbI_2$ precursor. Array indicates redshift after full iodine coordination. (**c**) Colloidal variation of the acid–base precursors prepared by stepwise addition of 25-μl $CH_3NH_2$ ethanol solution into 0.5 ml of 2M $HPbI_3$ (from left to right, 0 to 400 μl, white dash line lies after 300 μl). (**d**) Absorption spectrum variation of the acid–base precursors prepared by stepwise addition of 20-μl $CH_3NH_2$/ethanol solution into 0.3 ml of 1 M $HPbI_3$. (**e**) Absorption spectrum variation of 0.3 ml of 1 M control 1:1 $CH_3NH_3I:PbI_2$ precursors through stepwise addition of 20-μl $CH_3NH_2$/ethanol solution. The spikes at around 335 nm are caused by the precipitate after adding too much $CH_3NH_2$/ethanol solution.

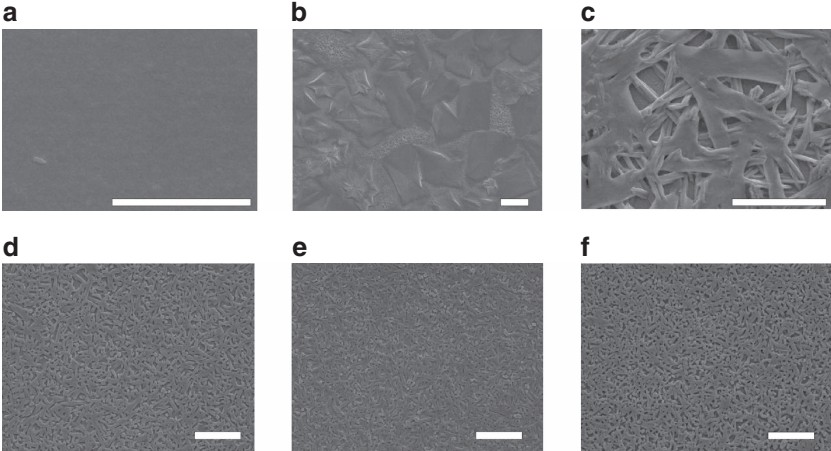

**Figure 3 | SEM of the thin films.** (**a**) The $PbI_2$ film. (**b**) $HPbI_3$ film. (**c**) Controlled perovskite film by one-step method. (**d-f**) NABR prepared perovskite films with increasing amounts of $CH_3NH_2$: 0.3 ml (**d**), 0.35 ml (**e**) and 0.4 ml (**f**) MA in 1.5 M 1 ml $HPbI_3$. Scale bar, 5 μm (**a**), 10 μm (**b-f**).

such peaks, which meant that DMF was still incorporated in the lattice (Fig. 4d). This information suggests that there exist iodine/ $CH_3NH_2$ vacant sites with larger binding energy in the control sample and thus DMF is not easily removed at room temperature. NABR method reduces the vacancies owing to full coordination of HI and $CH_3NH_2$ abundance. Namely, DMF molecules just play the role in solvation and/or intercalation through H-bonding to MA but are not directly coordinated to lead ions in NABR (Supplementary Fig. 1c). Hence, NABR facilitates quick transformation towards perovskite and affords crystalline perovskite even without baking (Fig. 4c and Supplementary Fig. 5a). Strong Bragg peaks of perovskite were observed at

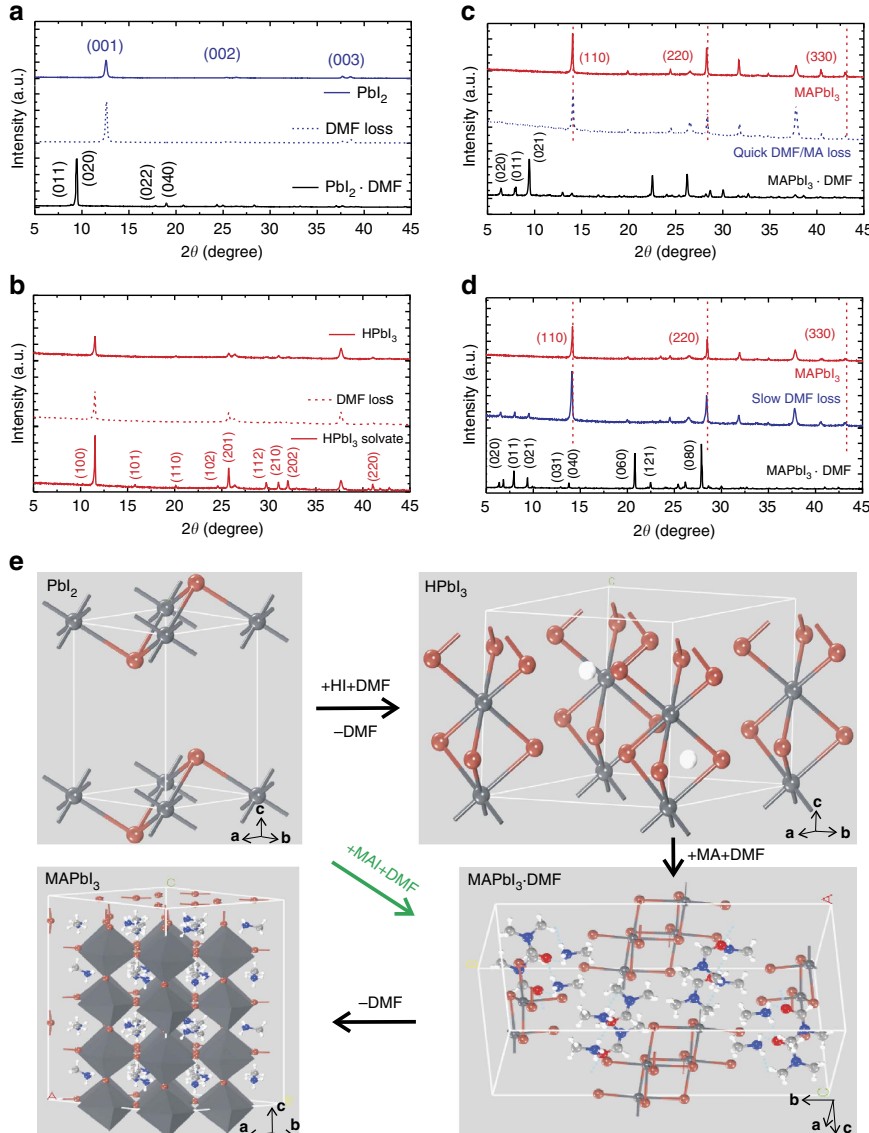

**Figure 4 | Crystalline phase conversion analysis. (a–d)** X-ray diffraction patterns of $PbI_2$ film (**a**), $HPbI_3$ film (**b**), $CH_3NH_3PbI_3$ films by NABR (**c**) and control one-step method (**d**) at different reaction stages represented by dip-coated wet film (bottom curve), spin-coated films before baking (middle curve) and after baking (top curve). (**e**) Crystallographic illustration between NABR conversion from $PbI_2$, to $HPbI_3$ ($H^+$ ions are included to indicate stoichiometry but are actually mobile around $[PbI_3]^-$ column), then to intermediate $CH_3NH_3PbI_3 \cdot DMF$, finally to $CH_3NH_3PbI_3$ and conventional conversion from $MAI + PbI_2$ using DMF as solvent, to $CH_3NH_3PbI_3 \cdot DMF$, then to $CH_3NH_3PbI_3$.

14.08°, 28.41° and 43.19° corresponding to the (110), (220) and (330) facets, respectively (Fig. 4c).

Figure 4e schematically illustrates the whole reaction process in conformity with crystallographic information. The Pb(II) centre in $PbI_2$ is coordinated by DMF, forming $PbI_2 \cdot DMF$ after spin-coating[28]. In the first-step of acid–base reaction, HI replaces DMF for direct coordination to Pb(II) in forming linear columns each composed of stacked face-sharing $PbI_6$ octahedra, which are further arranged to give a hexagonal array in $HPbI_3$ with the aid of DMF intercalation between them. In the second step, $CH_3NH_2/DMF$ is inserted into the inter-columnar region of $HPbI_3$, forming $CH_3NH_3PbI_3 \cdot DMF$ at first. The film morphology of perovskite is determined by the preformed monoclinic $CH_3NH_3PbI_3 \cdot DMF$ containing $[PbI_3]^-$ double chains that is thus needle-like (see Supplementary Fig. 1c). The $CH_3NH_3PbI_3 \cdot DMF$ opens its $[PbI_3]^-$ double chain after DMF removal and undergoes transformation to the tetragonal

perovskite structure. In traditional route, stoichiometric $MAI/PbI_2$ cannot ensure full iodine coordination for stoichiometric perovskite due to coordination competition by DMF at X sites.

Perovskite prepared by NABR has strong absorption in the green and weak absorption in the red, which is a feature of high-quality perovskite. The one-step method produces a weak flat light absorption spectrum probably due to the defect absorption[34] (Supplementary Fig. 5b). Hexagonal $HPbI_3$ has an absorption edge at 420 nm, different from the theoretical 0.3 eV band gap in its cubic perovskite phase[24], like $FAPbI_3$ with yellow and black phases[35].

**Intrinsic stability of as-prepared films.** In principle, $PbI_2$ component is difficult to dissolve in water and thus the unstable component is the organic $CH_3NH_3I$ counterpart. In general, a 2D

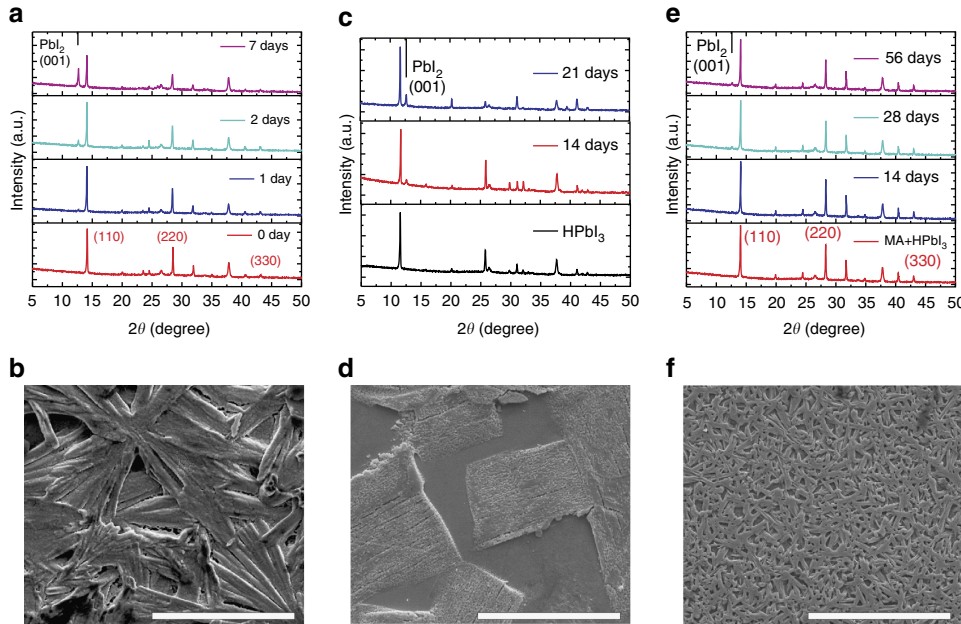

**Figure 5 | Humidity stability of as-prepared film.** (**a**) X-ray diffraction patterns of control perovskite film before and during degradation. (**b**) SEM micrograph of control perovskite film after 7 days degradation. (**c**) X-ray diffraction patterns of HPbI$_3$ before and during degradation. (**d**) SEM micrograph of HPbI$_3$ after 21 days degradation. (**e**) X-ray diffraction patterns of perovsky by NABR before and during degradation. (**f**) SEM micrograph of perovskite by NABR after 56 days degradation. Notes: the films were deposited on FTO glass and exposed to 65% humidity in a cabinet containing water in a beaker. Scale bar, 30 μm.

perovskite bearing a long alkyl chain for shielding from moisture has much higher stability. For CH$_3$NH$_2$, the −CH$_3$ group is hydrophobic and thus good humidity stability could be expected if -NH$_3$ is well bonded to inorganic [PbI$_3$] framework.

We have monitored the degradation of these perovskites (Fig. 5) in ~65% humidity and found that the control perovskite prepared by the one-step method was severely jeopardized by moisture after 1 week (Fig. 5a). The final products contained large amounts of PbI$_2$ as evidenced by PbI$_2$ (001) peak at 12.6° (Fig. 5a), together with amorphous solvate judging from the morphology (Fig. 5b). The perovskite prepared by traditional two-step method change to yellow for about 2 weeks under the same condition (Supplementary Fig. 3e), and mixed CH$_3$NH$_3$PbI$_{3-x}$Cl$_x$ perovskite degraded quickly within 1 h in such humidity to CH$_3$NH$_3$PbI$_{3-x}$Cl$_x$·H$_2$O (Supplementary Fig. 1).

We have also checked the stability of starting HPbI$_3$. HPbI$_3$ displayed much better stability than traditional perovskites after 3 weeks, judging from the latter appearance of PbI$_2$ (001) peak (Fig. 5c). Except for some erosion traces on the surface, we did not observe too much change in SEM (Fig. 5d). It has to be mentioned that the HPbI$_3$ directly precipitated by adding anti-solvents (such as diethyl ether) does not endure high humidity, probably due to HI/H$_2$O residues.

The perovskite prepared by NABR had robust stability after optimizing the amount of CH$_3$NH$_2$ in this work. The best film remained stable for about 2 months in ~65% humidity, without distinct PbI$_2$-impurity from X-ray diffraction pattern and significant morphology change from freshly prepared samples (Fig. 5e,f). However, we found that stability was CH$_3$NH$_2$-amount-dependent (Supplementary Figs 6–12). The CH$_3$NH$_2$ used should be largely in excess in NABR, and CH$_3$NH$_2$ deficiency leads to incomplete conversion of HPbI$_3$ with poor stability (Supplementary Fig. 6). Therefore, we confirm that HPbI$_3$ permits the formation of a stable well-defined perovskite framework, and excess CH$_3$NH$_2$ ensures sufficient

filling in the lattice and surface passivation to resist H$_2$O erosion in combination.

We have further performed TGA to check the thermal stability of perovskite films and starting materials. TGA curve for CH$_3$NH$_3$I shows nearly 100% weight loss between 260 and 320 °C (Supplementary Fig. 13). HPbI$_3$ has a large weight loss at a temperature range between 300 and 360 °C, which is indicative of the release of HI. Consistent with its low humidity stability, the perovskite prepared by one-step method was also not thermally stable. Weight loss onset occurred at 60 °C, which was consistent with a previous report[22] and meant that the organic–inorganic components were not tightly bounded. Through careful observation, the sequential thermal decomposition mechanism could be identified for control perovskite (see reaction 10) judging from two different weight loss regions at 60–150 °C and 250–350 °C, with the similar weight loss at 250–350 °C to HPbI$_3$. For NABR, the weight loss of the organic component was much larger than the others, being indicative of a fully coordinated [PbI$_3$]$^-$ scaffold with sufficient CH$_3$NH$_2$ filling in the perovskite lattice.

$$CH_3NH_3PbI_3 \xrightarrow{60-150\,°C} HPbI_3 + CH_3NH_2 \xrightarrow{250-300\,°C} PbI_2 + HI \tag{10}$$

Probably, perovskite prepared by the two-step method and NABR also decomposed sequentially under heating stress in terms of thermodynamics. Due to the similar release rates of MA and HI in kinetic pathways, we were unable to detect sequential events. In principle, the sequential decomposition thermodynamics is acceptable because when iodide is well coordinated to the metal centre of PbI$_2$, and then the bind energy of MAI is reduced, leading to releasing MA easily.

TGA was performed in the low-temperature region for release of the organic component, with 30 and 120 min heat preservation at 100 and 200 °C for clear observation of weight

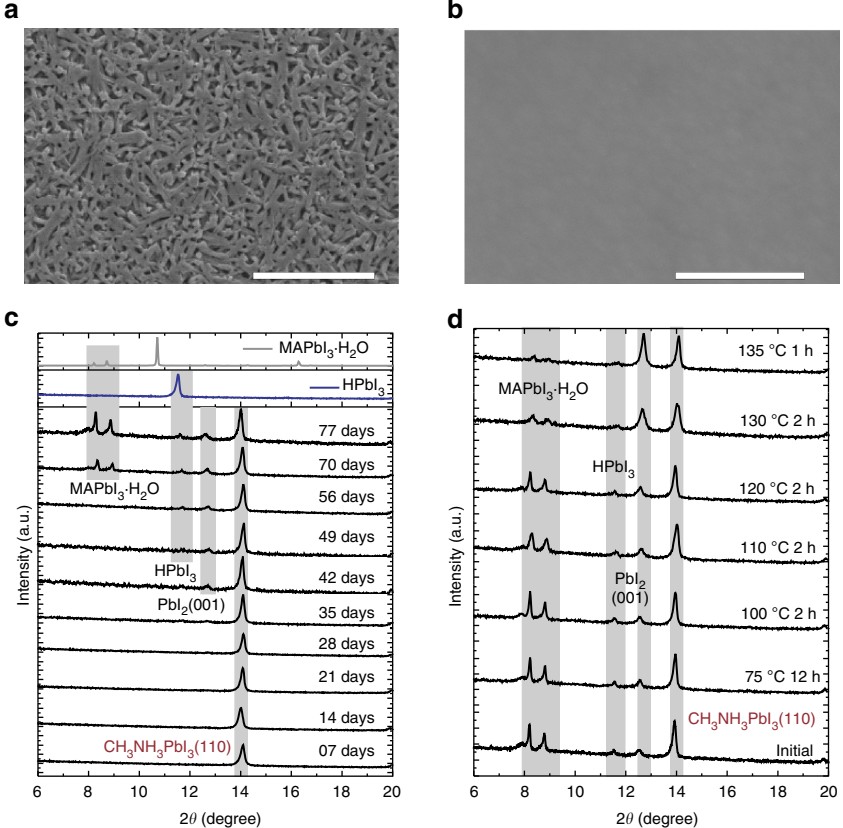

**Figure 6 | Pin-hole-free film and stability.** (**a**,**b**) Perovskite film coverage using nucleation agent (NA) (**b**) and without NA (**a**) by NABR. Scale bars,10 μm (**a**) and 0.5 μm (**b**). (**c**) Further stability check for optimized film with even longer exposure times over 2 months. (**d**) Reversible test of degraded film through heating.

loss, respectively. We can see the better stability in NABR after optimization (Supplementary Fig. 13b). Besides, NABR affords to endure high-temperature calcination below 150 °C for high-quality films with negligible impurity (Supplementary Fig. 14). Therefore, NABR represents a controllable way towards the preparation of highly stable perovskite in humid environment and under heat stress.

**Film optimization and stability check**. We have introduced dripping of nucleation agent during spin-coating, which enhanced the film coverage for photovoltaics. The nucleation agent made of anti-solvent increased the heterogeneous nucleation sites and reduced the height of the free-energy barrier ($\Delta G$) for nucleation. Therefore, it accelerated nucleation exponentially according to classical nucleation theory ($R = Ke^{-\Delta G/k_B T}$, $R$, nucleation rate; $K$, constant; $k_B T$ thermal energy) and suppressed the Ostwald ripening in sequential nucleation, resulting in 100% film coverage by this key step (Fig. 6a,b, and Supplementary Figs 15–17). Through nucleation control, we could obtain pin-hole-free thin-film with optimization (Fig. 6b).

We have found that optimized perovskite thin-film with nucleation agent remained stable through colour and X-ray diffraction monitoring (Fig. 6c). Under ∼65% humidity, there was no signature of PbI2-impurity after 1 month of exposure and negligible PbI2 (∼7%) from X-ray diffraction monitoring. However, degradation beyond 2 months of the pin-hole-free film was quite different from that of the mesoporous as-prepared film without nucleation agent. We found a plausible trace of HPbI3 after 7 weeks at $2\theta = 11.6°$, and then peaks at $2\theta = 8.1°$ and 8.7° indicating (001), (100) facets of $CH_3NH_3PbI_3 \cdot H_2O$ after 9 weeks, respectively. This can be explained by the sequential degradation

thermodynamics in combination with degradation kinetics if we consider the surface effect. On the surface of pin-hole-free film, due to the ready release of both HI and $CH_3NH_2$, the sequential degradation reaction (5–7) occurred nearly simultaneously and thus only yielded PbI2 that we could observe. When degradation went to the interior that was single crystal-like[36], $CH_3NH_2$ and HI were difficult to be released from the interior of high-quality of the film, thus as formed internal HPbI3 and $CH_3NH_3PbI_3 \cdot H_2O$ were detected by X-ray diffraction patterns after long-time degradation. This explanation has been further tested by the following experiments. First, we tried to convert the $CH_3NH_3PbI_3 \cdot H_2O$ back to $CH_3NH_3PbI_3$ through heating (Fig. 6d). To our surprise, after 12 h heating at 75 °C and 2 h heating at 100 °C, we found there was no observable change of the monohydrated phase. We then increased the calcination temperature gradually and found a change happened at above 110 °C, which was actually the decomposition threshold of defective perovskite. Second, we also checked the morphology after 70 days degradation and found that the rod-like degradation products were really embedded in the film (compare Supplementary Fig. 18 with Fig. 5b), which thus confirmed our assumption. Finally, we once again draw attention to degradation of the perovskite film without nucleation agent above. Due to the mesoporous structure that was fully exposed to moisture, we did not observe the transition products as the bulk-like pin-hole-free film.

**Performance evaluation**. The PV performance for freshly prepared and humidity-exposed films on planar compact TiO2 and mesoscopic TiO2 was carefully conducted in this work (see the Methods section, Fig. 7a, Supplementary Figs 19–22 and

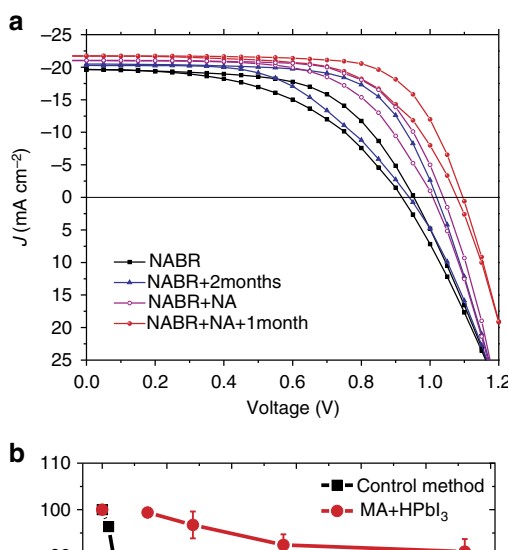

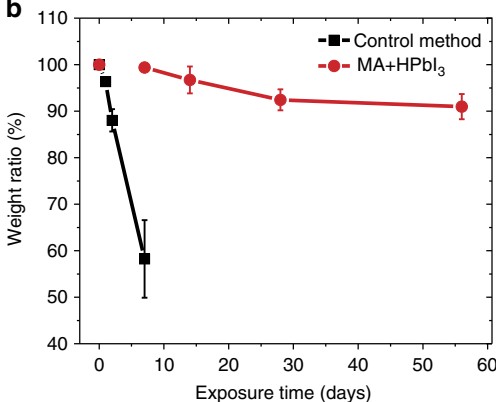

**Figure 7 | Photovoltaic optimization.** (**a**) Optimized solar cell performance with NA (pink line) and without NA (black line) and corresponding solar cell performance after moisture exposure with NA (blue line) and without NA (red line). Notes: perovskite thin films are exposed to moisture and then fabricated as devices. (**b**) Quantitative estimation of degradation through X-ray diffraction external standard method (65% moisture with ambient light soaking). Error bars represent s.d. calculated from five thin films prepared at the same conditions.

Supplementary Tables 1–4). Without nucleation agent, the solar cell displayed low performance due to the poor coverage. The humidity-exposed films had much better power conversion efficiency (14.0%) than freshly prepared film (11.1%) in this system, which meant the robust stability of NABR produced perovskite. In detail, the freshly prepared film produced $V_{oc} = 0.90$ V, $J_{sc} = 19.6$ mA cm$^{-2}$, $FF = 0.520$ and overall $PCE = 9.1\%$ in forward scan, and $V_{oc} = 0.96$ V, $J_{sc} = 19.6$ mA cm$^{-2}$, $FF = 0.590$, overall $PCE = 11.1\%$ in reverse scan. After humidity exposure, it produced $V_{oc} = 0.94$ V, $J_{sc} = 20.3$ mA cm$^{-2}$, $FF = 0.541$, and overall $PCE = 10.3\%$ in forward scan, and $V_{oc} = 1.01$ V, $J_{sc} = 20.4$ mA cm$^{-2}$, $FF = 0.681$, overall $PCE = 14.0\%$ in reverse scan (Fig. 7a).

After adding nucleation agent, the solar cell displayed much higher performance due to the improved film coverage after series of device optimization (Fig. 7a). The freshly prepared film produced $V_{oc} = 1.05$ V, $J_{sc} = 21.1$ mA cm$^{-2}$, $FF = 0.684$ and overall $PCE = 15.1\%$ in reverse scan. After 1 month humidity exposure, it produced $V_{oc} = 1.08$ and 1.07 V, $J_{sc} = 21.7$ and 21.7 mA cm$^{-2}$, $FF = 0.725$ and 0.651, and overall $PCE = 17.0$ and 15.2% in reverse/forward scans. The average 16.1% $PCE$ is among the highest efficiency PSC using stable material. This improved performance after humidity exposure demonstrated the improved stability as well.

The improved performance suggests that a thimbleful amount of $H_2O$ is beneficial to solar cell efficiency. We have tried to track

the doping effect and/or surface passivation on solar performance and quantitatively characterize the concentration of pure perovskite film using external standard method with X-ray diffraction patterns (Fig. 7b). The weight concentration was characterized for the control film and NABR produced film. We can see that over 90% perovskite remained after about 2 months, while only about 60% perovskite residue was found after 7 days degradation from the control method. The best performance of perovskite against humidity has doping concentration of about 7%, which is generally consistent with the concentration range in previous work using PbI$_2$-rich inorganic/organic composition[37].

Besides, it is notable that stable PSCs through layer shielding reported in previous works are generally lower than 16.2% efficiency in reverse scan[20,38]. According to the detailed reports, MAPbI$_3$ perovskite itself could not endure high humidity (55%) for 1–6 days without encapsulation of its high-efficiency photovoltaics, which indicates that its high stability primarily comes from device encapsulation. However, here we demonstrate that NABR using excess CH$_3$NH$_2$ to react with well-defined HPbI$_3$ provides a reliable route to producing material-stable perovskites superior than those reported in previous work[20,38].

This alternative NABR method is probably beneficial for long-term development and large-scale production under ambient conditions, thus reducing the cost in device fabrication and encapsulation. First, in NABR, both CH$_3$NH$_2$ and HPbI$_3$ are chemically stable and thus readily facilitate fabrication. Moreover, the synthesis of HPbI$_3$ is much easier than CH$_3$NH$_3$I and can be collected using ethanol. Second, fundamentally speaking, the use of well-defined HPbI$_3$ that is stoichiometrically identical to the intermediate complex as starting material is more chemically reasonable. Besides, we have fabricated a large area PSC up to $1.0 \times 0.5$ cm$^2$ in area, which successfully delivered 15.0% $PCE$ at the present stage. Thus, the present study demonstrates up-scaling potential for the assembly of modules in solar cells that function efficiently under ambient condition (Supplementary Fig. 23, Supplementary Table 5).

## Discussion

In conclusion, we have investigated the degradation and recovery of CH$_3$NH$_3$PbI$_3$ perovskite and established an improved stability process. The degraded perovskite can be recovered as fresh perovskite using methylamine CH$_3$NH$_2$, which means that methylamine can substantially retard the degradation. On the basis of this understanding, we have developed an alternative route using NABR to facilitate the synthesis of perovskite. This NABR procedure, involving the production of HPbI$_3$ using excess HI acid and PbI$_2$ as well as subsequent reaction between excess CH$_3$NH$_2$ base and HPbI$_3$ acid, provides CH$_3$NH$_3$PbI$_3$ perovskite thin-film that is highly stable under ~65% humidity for 2 months without appreciable PbI$_2$-impurity, whereas other perovskites prepared by the traditional one-step and two-step methods withstand degradation <1 week and 2 weeks, respectively. We have identified a high-quality form of HPbI$_3$ with identical Pb(II) coordination number to perovskite and CH$_3$NH$_2$ abundance as two important factors towards stable perovskite with as less site vacancies as possible. Excess and volatile acid/base leads to full coordination and stoichiometry, respectively, thus eliminating the penetration of water vapour and improving the stability in highly humid environments. The device has been optimized to 17.0/15.2% $PCE$s in forward/reverse scans after 1 month exposure in ~65% humidity. This work provides an important insight into intrinsic stability and efficiency of perovskite as well as the utilization of simple reaction procedure with up-scaling potential via bottom-up synthetic chemistry for high-performance photo-

voltaics. Through the reaction demonstration, the vacancy-free insight into improving stability is probably a general paradigm for other perovskites. For example, $FA_xCs_{1-x}PbI_yBr_{3-y}$, in which the $FA^+/Cs^+$ is also difficult to be removed, delivers as less vacancy as possible at the A sites of $AMX_3$ perovskite for high stability. It is challenging to make $MAPbI_3$ stable in traditional routes so far. Our NABR route provides a way to enhance the stability through smart reaction control to reduce vacancy without any other material composition. The intermediate structure of $HPbI_3$ is especially interesting, in which the proton seems to move freely around the $[PbI_3]$ column. The perovskite after humidity test has even higher performance in this work, which may arise from protons in $HPbI_3$.

## Methods

**Materials.** Methylammonium iodide (MAI, Dyesol), $PbI_2$ (Sigma-Aldrich, 99%), N,N-dimethylformamide (DMF, Sigma-Aldrich, anhydrous, 99.8%), spiro-OMe-TAD (Merck), 4-tert-butylpyridine (Sigma-Aldrich, 96%), Titania paste ($TiO_2$, 30 nm, Dyesol), Titanium(IV) isopropoxide (Sigma-Aldrich, 99.999%), methylamine (Sigma-Aldrich, 33% in ethanol), hydroiodic acid (HI, 57% in water), lithium bistrifluoromethanesulfonimidate (LiTFSI, Sigma-Aldrich, 99.95%), chlorobenzene (Sigma-Aldrich, anhydrous, 99.8%), and all other chemicals were used as received without further purification.

$HPbI_3$ preparation: $HPbI_3$ powder was prepared by mixing $PbI_2$ and excess HI (1.5:1 molar HI:$PbI_2$) in DMF to ensure complete conversion, and stirring at 40 °C overnight. The light yellow precipitates were obtained by washing the precursor in abundant ethanol to remove excess HI until the supernatant turned to white. The excess HI and ethanol were then removed through filtration. The resulting powders were further dried and stored in an oven at 60 °C. It was then re-dissolved in DMF solution and different amounts of $CH_3NH_2$ ethanol solution were freshly added to obtain the NABR perovskite precursors. The resulting $HPbI_3$ sample washed by diethyl ether did not exhibit high stability according to our control experiment. Needle-shaped semi-transparent $HPbI_3$ solvate single crystals were grown by dissolving 1.5:1 molar HI:$PbI_2$ in DMF followed by vapour diffusion of chlorobenzene into the mother liquor at 80 °C for 6 hours, yielding $HPbI_3$ solvate crystal. After heating at 80 °C for 30 min, it turned faint yellow quickly and converted to $HPbI_3$ crystal according to single crystal X-ray diffraction. The weight (HI + solvent):$PbI_2$ ratio was 37.8% judged from TGA at 385 °C for single-crystal $HPbI_3$ solvate, which was freshly collected from the mother liquor and placed on filter paper for 5 min, probably suggesting chemical formula $HPbI_3 \cdot xDMF$ ($x \leq 1$) with disordered DMF inside the lattice (Supplementary Figs 4 and 24). The quick weight loss before 60 °C suggests the easy release of DMF in $HPbI_3 \cdot xDMF$ even at room temperature, yielding pure $HPbI_3$ indicated by 28% weight ratio at 385 °C (HI/$PbI_2$) in weight loss after 60 °C. The $HPbI_3$ powder recorded a 27.8% HI/$PbI_2$ weight ratio at 350 °C by TGA (Supplementary Fig. 13), which confirms the molecular formula $HPbI_3$.

**Crystal structure analysis.** X-ray intensities of $HPbI_3$ solvate and $HPbI_3$ were collected at 296 K on a Bruker AXS Kappa Apex II Duo diffractometer with $MoK_\alpha$ radiation ($\lambda = 0.71073$ Å) from a sealed-tube generator. Crystal data: hexagonal, $a = b = 8.7517$ Å, 8.7339 Å, $c = 8.1802$ Å, 8.1770 Å for $HPbI_3$ solvate and $HPbI_3$ with 1:3 atomic Pb/I ratio, respectively. The systematic absences are consistent with both noncentric space group P63mc (No. 186) and centric space group P63mcm (No. 193). As the proton has negligible X-ray scattering and the DMF molecule exhibits severe orientational disorder, structure determination was based on space group P63mc with only the Pb and I atoms subjected to anisotropic least-squares refinement using the SHELXL-97 program; 7,944 reflections measured, of which 509 are unique, $R_{int} = 0.099$, $R1 = 0.044$, $wR2 = 0.134$ and $GOF = 1.05$. The crystallographic data for this paper have been deposited with the Cambridge Crystallographic Data Centre (CCDC) as No.1479488. These data can be obtained free of charge from CCDC via www.ccdc.cam.ac.uk/data_request/cif and supporting crystal information files (Supplementary Table 6).

**Device preparation.** F-doped $SnO_2$ (FTO) (TEC08) substrates were cleaned in an ultrasonic bath sequentially with acetone, 2-propanol and ethanol for 15 min, separately. The $TiO_2$ precursor was prepared by 0.6 ml titanium isopropoxide and 0.15 ml 37%w/w HCl solution dissolved in 15 ml ethanol. The dense blocking layer $TiO_2$ was coated onto FTO substrate by spin-coating of titanium precursor at 5,000 r.p.m. for 40 s, followed by annealing in air at 500 °C for 30 min. A 14 wt% solution of $TiO_2$ nanoparticles in ethanol was spin-coated onto the dense $TiO_2$ layer at 5,000 r.p.m. for 40 s to form a mesoporous scaffold and sintered in air at 550 °C for 30 min. After cooling to room temperature, the mesoporous $TiO_2$ was immersed in aqueous 30 mM $TiCl_4$ at 70 °C for 30 min, rinsed with DI water and annealed at 500 °C for 20 min. The perovskite precursor was prepared by mixing equimolar ratio of MAI and $PbI_2$ in DMF by stirring at 60 °C. This solution was then spin-cast onto the $TiO_2$ films at 3,000 r.p.m. for 30 s and annealed at 100 °C for 15 min to form the control $CH_3NH_3PbI_3$ samples. For the perovskite precursors formed by $HPbI_3$, 33 wt% MA solution in ethanol was mixed with 1.5 M $HPbI_3$

DMF solution with increased volume of MA, and the fresh precursor was spin-coated on $TiO_2$ at 2,500 r.p.m. and annealed at 100 °C for 30 min. With device optimization, we also used other concentration $HPbI_3$ in association with excess MA and introduced nucleation agent (toluene, 1 ml) during the spin-coating at 10 s for full coverage. The spiro-OMeTAD was prepared by dissolving 75 mg spiro-OMeTAD with 25 μl LiTFSI solution (520 mg in 1 ml acetonitrile) and 35 μl 4-tert-butylpyridine in 1 ml chlorobenzene. Hole transport layer was deposited on the annealed perovskite film by spin-coating at 5,000 r.p.m. for 40 s to get the optimized efficiency. The devices were placed in a moisture-controlled cabinet overnight for oxidization of spiro-OMeTAD. Finally, 80 nm Au electrode was deposited by thermal evaporation with shadow mask area of 0.1 cm². All device fabrication procedures were carried out in $N_2$-purged glovebox.

**Stability experiments.** For all the humidity stability studies, the perovskite films were spin-coated on FTO glasses, and the films were stored at room temperature (measured as 25 ± 1 °C) in a controlled-humidity cabinet with a transparent glass door, with a low-power fluorescent light in the room. The relative humidity in the chambers was controlled to the desired humidity 65 ± 5% by a beaker of water. And the air humidity in the room is floating from 60 to 75%. The relative humidity in the cabinet was precisely measured periodically throughout the stability experiments using a calibrated hygrometer. The cabinet was only opened just when taking sample out. As the humidity difference is very small, and the humidity cannot be greatly affected when opening the cabinet door and return to the desired humidity by the beaker of water through measurement. To emphasize the stability of perovskite itself and check the performance after humidity exposure, we deposited perovskite films without and with nucleation agent during spin-coating and then expose them in ∼65% humidity for 1–2 months. Humidity-exposed films were assembled into solar cells in the configuration of FTO/$TiO_2$/perovskite/spiro-OMeTAD/Au and compared with freshly prepared films.

**Basic characterization.** The extinction and absorption spectra of solution samples were measured on a Hitachi U-3501 ultraviolet/visible/NIR spectrophotometer. The general images of the film morphology were obtained using an FEI Quanta 400 field emission scanning electron microscope (FESEM, FEI, Quanta 400 FEG) operated at 10 keV. X-ray diffraction measurements were performed with a Bruker D8 Advance Davinci powder X-ray diffractometer using a $CuK_\alpha$ source. TEM imaging was performed on an FEI Tecnai Spirit microscope operating at 120 kV.

**Solar cell test.** The current density-voltage curves of solar cells were measured (Keithley Instruments, 2612 Series Source Meter) under simulated AM 1.5 sunlight generated by a 94011A-ES Sol series Solar Simulator. The solar cell devices were tested in $N_2$-filled glovebox under room temperature. Solar cell performance was scanned at scan speed 0.5 V s⁻¹, dwell time 0.1 s, voltage step 0.05 V in forward and reverse scan loop. The scanning parameters related to the performance were attached (Supplementary Fig. 25). We used the 'Nicht abdecken' sensor for the light source checking and then measured the devices. The effective solar cell area was defined by the shadow mask as 0.1 cm².

**Data availability.** The authors declare that the data that support the findings of this study are available from the corresponding author on reasonable request.

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

## Acknowledgements

The work is in part supported by Research Grants Council and Innovation and Technology Fund of Hong Kong, particularly via Grant Nos T23-407/13-N, CUHK 14204616, PolyU 252001/14E, ITS/004/14, CUHK Group Research Scheme and National Natural Science Foundation of China (Grant Nos.61674070). We thank Dr Zhu Houyu and Professor Guo Wenyue in China University of Petroleum for some theoretical insights.

## Author contributions

K.Y.Y. and J.B.X. conceived and supervised the project. M.L., T.Z. and K.Y. performed the experiments. C.F.N. and T.C.W.M. did the single-crystal X-ray diffraction and verified the structure of HPbI₃. K.Y., Y.C. and J.X. analysed the data. K.Y. built the material model and wrote the manuscript. T.C.W.M. and K.Y. revised the manuscript. All the authors discussed the results and commented on the manuscript.

## Additional information

**Competing financial interests:** The authors declare no competing financial interests.

**Publisher's note**: 

