## [Peer Review File · Nature Communications]

Reviewer #1 (Remarks to the Author)

The manuscript by Mingzhu Long et. al. describes an interesting route to prepare perovskite thin films that are stable for at least two months. This is an important advance in the stability of perovskite solar cells, especially when one considers the possibility of combining the process with cell encapsulation. Stability of perovskites is the major issue for the future viability of perovskite-based optoelectronic devices. Therefore, the route described here should be of broad interest to the perovskite community. I recommend publication of this paper after some minor corrections.

- HPbI₃ was used as a new precursor in DOI: 10.1002/adfm.201404007. Therefore, the authors should explain the difference between their work and the published one.

- Page 1: "finally to PbI₂ and HI vapor" should be rephrased to "finally PbI₂ solid and HI vapor".

- Authors attribute the enhanced stability of their perovskite films to the full coordination of PbI₃-framework. Could the authors comment on whether the better stability of perovskite single crystals (DOI: 10.1038/ncomms9724) is also related to this phenomenon? Since single crystals are usually closer to the stoichiometric fully coordinated compound than typical thin films.

- Fig 1e,f scale bar should be given and how those crystals were prepared.

- The authors found that their devices displayed even better performances after being exposed to moisture for 1 month. This finding is very interesting and some comment from the authors on this point would improve the impact of the paper.

Reviewer #2 (Remarks to the Author)

A. This article presents an alternative synthesis route to make the Organic-Perovskite based photovoltaics that show better moisture resistance. Currently one of the challenges in commercialization of this highly efficient material is its degradation when it is exposed to moisture. An acid based reaction that produces HPbI₃ through excess HI reacting with PbI₂ and subsequent reaction between CH₃NH₂ and HPbI₃ is shown to deliver thin films that are stable in ~65% humidity for up to two months. The results are verified using thin film XRD, TEM, TGA and UV measurements.

B. The work is reasonably original. Some literature search reveals that this method of synthesis has not been tried before. However, I am not convinced that this is of great general interest as to be published in nature communications. The other thing to mention here is there is negligible formation of PbI₂ impurity after two months, however what happens after two months? Is this time sufficient for commercialization or is more work required to lengthen this time period. The argument presented is that two months is significantly longer than previous work which showed stability for weeks. There are other work referenced here which can accomplish similar stability by cell design [ref 20]. Why is that not the correct approach for the fabrication of cells? Would this process suggested here be more cost effective? Is the synthesis scalable to large production?

Given the study is motivated by large scale production, some of these discussion should be included to strengthen the novelty and application.

C. Most of the data is used in more of a qualitative than quantitative analysis. Often phase identification is done on the basis of one reflection. Since the work done is on thin film, the crystallographic models can only be guessed but a full model refinement on bulk would be more convincing for the synthesis mechanism suggested. There is no mention of the film thickness which can be important for application. So most of what is being discussed is surface effects. However to confirm the hypothesis in a stronger platform would need looking at the bulk as well. The presentation of the material is reasonable, however, a large number of acronyms have been used without first spelling them out (i.e. DMG, PL, PSC).

D. As mentioned earlier, very little data fitting has been applied and most of the data is used in a qualitative manner so not much statistical treatment is presented in the paper.

E. Conclusions: Conclusions are supported by the observation presented.

F. Page two, the sentence "Based on previous work by others....." needs references

Equation (2-i) does not make sense. How can $A + B = B - A$? Also given the crux of the paper is a new synthesis mechanism, clearly one would like to know what is the final yield of the target product?

Throughout the document the small angle has been wrongly used. When one talks about small angle in x-ray, 2 □□□ In the sca
recommended that this is presented in natural units for structural measurement which is generally d-spacing.

G. See F. The references are appropriate and give credit to previous work.

H. As mentioned earlier, the introduction needs to discuss about scale up and cost effectiveness, given the main argument here is this is needed for commercialization of solar cells made with these materials.

Reviewer #3 (Remarks to the Author)

In the presented paper, degradation of the methylammonium lead iodide is briefly analyzed to propose the facile two-step non-stoichiometric acid-base route of its thin film synthesis. Colloidal stability of precursors (HPbI₃, MA+HPbI₃, and control MAI+PbI₂) studied as well as the stability of materials in time and under humidity. Authors propose HPbI₃ as an intermediate compound. The significance of the MAPbI₃ thin film formation study is important due to the rising issue of a long-term stability of PV devices based on perovskite materials. Authors mainly are trying to solve the problem of volume-expansion control during the two-step synthesis and the obtained perovskite is appeared to resist degradation judging from XRD patterns measured after 56 days. The paper is novel and interesting because it brings to light the role of HI excess during the first step and the influence of the synthetic route on the morphology of the thin film. Still a few inconsistencies can be noticed. First, authors quickly jump to the conclusion that HPbI₃ is the intermediate step during the synthesis, whereas many authors have been more cautious to make such prediction. Though there have been evidences of the enhanced coordination in the HI + PbI₂ + DMF solution in the presented paper (colloidal behavior of the latter solution; red-shift in the absorbance spectrum; expansion of the lattice spacing comparing to PbI₂ in the deposited thin

film), the exact mechanism still remains unclear. It may be that the role of the solvent DMF and its intercalation between PbI₂ sheets is underestimated here. Nevertheless, there is no doubt that the I-coordination occurs in HI-rich solutions of PbI₂ and the paper opens the prospect for the further investigations (as well as investigation of the structure of these intermediate complex and the role of protone). In addition, recently HPbI₃ was also mentioned elsewhere[1], denoting that the role of polyplumbate complexes as an intermediate step in the synthesis of perovskites might be of a certain interest.

Second, the degradation mechanism presented in the paper is highly simplified paying attention to the fact that the recent crystallographic study[2] revealed formation of monohydrate and more complex dehydrate during the decomposition under humidity. The oversimplification may be the main issue leading to misinterpretations.

Moreover, each of the two-step reactions of the acid-base precursor preparation are proposed to be exothermic without no significant evidences or references.

In addition, few technical flaws have to be mentioned. Among them are sporadically used incorrect phrasal constructions (e.g. "...until we saw the precipitation in the naked eyes..."), which do not facilitate the quick understanding of the text. Also, the element "Figure 1 (e,f)" depicting degraded perovskite crystals does not have a scale to compare the real size. The same remark applies to a few figures in the Supplementary (S7, S19).

Besides small flaws and too generalized conclusions, technical side of the paper remains on the high level and the topic of the paper is important for the investigation of the perovskite stability in terms of a large-scale manufacturing. Authors managed to show a remarkable PV performance of the constructed devices and the stability of the perovskite material.

[1] Christians, Jeffrey A., Pierre A. Miranda Herrera, and Prashant V. Kamat. "Transformation of the excited state and photovoltaic efficiency of CH₃NH₃PbI₃ perovskite upon controlled exposure to humidified air." *Journal of the American Chemical Society* 137.4 (2015): 1530-1538.

[2] Wang, Feng, et al. "HPbI₃: A New Precursor Compound for Highly Efficient Solution-Processed Perovskite Solar Cells." *Advanced Functional Materials* 25.7 (2015): 1120-1126.

Response to Reviewers' comments:

Reviewer #1 (Remarks to the Author):

The manuscript by Mingzhu Long *et al.* describes an interesting route to prepare perovskite thin films that are stable for at least two months. This is an important advance in the stability of perovskite solar cells, especially when one considers the possibility of combining the process with cell encapsulation. Stability of perovskites is the major issue for the future viability of perovskite-based optoelectronic devices. Therefore, the route described here should be of broad interest to the perovskite community. I recommend publication of this paper after some minor corrections.

- HPbI₃ was used as a new precursor in DOI: 10.1002/adfm.201404007. Therefore, the authors should explain the difference between their work and the published one.

Answer: We thank this comment aiming to enhance the novelty of our synthetic route. The HPbI₃ crystallinity was well-defined and stoichiometrically (PbI₃) identical to perovskite through nonstoichiometric reaction between excess HI and PbI₂ (~1.5:1), which referred to the work by Wang *et al.* (Our research teammate), with careful modification using ethanol to wash the products instead of diethyl ether. The alcohol washing refines the HPbI₃ through the removal of the excess HI residue that will produce MAI during MA exposure, and thus ensures the quality of HPbI₃, leading to improved stability of both HPbI₃ and MAPbI₃. The HPbI₃ washed by diethyl ether can't deliver such high stability.

Second, although the reaction between 1:1 FAI:HPbI₃ mixture produced FAPbI₃ at above 150 °C,

the mixture precursor with 1:1 HPbI₃:MAI mixture did not produce MAPbI₃ (see the Figure below). The FAI+HPbI₃ reaction was actually decomposition of FAI into FA+HI at above 150°C, with subsequent reaction between FA+HPbI₃ for FAPbI₃. Such high temperature will decompose MAPbI₃ perovskite thus does not allow to yield perovskite using MAI+HPbI₃ (decomposition rate>yield rate), which meant MA+HPbI₃ reaction was the optimum reaction path (Produce perovskite after spin-coating without heating). Therefore, we address the reaction pathway related to intermediate reaction for perovskite formation, which points out another novelty of this work in the reaction chemistry.

See revisions in Page 3 in below the sub-headline “acid-base reaction” or the following:

“**Acid-Base Reaction.** In order to synthesize stable perovskite, we propose a two-step route through NABR based on above analysis.

In reaction (3), the excess hydroiodic acid promotes the reaction completely and yields HPbI₃ that is stoichiometrically identical to [PbI₃]⁻ framework of perovskite, permitting to form fully coordinated framework for perovskite without I-vacancy. In reaction (4), the excess CH₃NH₂ allows the full reconversion of HPbI₃ to perovskite and thus eliminated the CH₃NH₂ vacancy. Different from use of excess CH₃NH₃I reacting with PbI₂, both CH₃NH₂ and HI are facile to remove towards stoichiometric lattice due to the volatility at

room temperature. Besides, the two reactions are probably exothermic due to acid-base neutralization and could occur spontaneously.

The starting material of HPbI₃ was prepared referring to previous work³¹, with careful modification using ethanol instead of diethyl ether to remove excess HI and precipitate products, followed by air pump filtration and drying at 60 °C overnight for purification. The HPbI₃ washed by diethyl ether cannot deliver high stability according to our control experiment.”

- Page 1: "finally to PbI₂ and HI vapor" should be rephrased to "finally PbI₂ solid and HI vapor".

Answer: The correction has been made with appreciation.

- Authors attribute the enhanced stability of their perovskite films to the full coordination of PbI₃- framework. Could the authors comment on whether the better stability of perovskite single crystals (DOI: 10.1038/ncomms9724) is also related to this phenomenon? Since single crystals are usually closer to the stoichiometric fully coordinated compound than typical thin films.

Answer: We thank the referee's suggestion. Indeed, the stoichiometric composition can enhance the stability. As known, the large crystal has better stoichiometric ratio due to small surface to volume ratio and thus the large crystal has better stability than small crystal (Fig. S2b). Therefore, we would like to comment it with comparison. Besides, even for large crystals, we have to mention that perovskite microcrystals prepared by traditional methods are not as stable as perovskite thin film prepared by our NABR method (Fig. S2).

See Page 5 or below.

“We have found that perovskite remained stable through color and XRD monitoring. (Fig. 5b) Under ~65% moisture, there was no signature of PbI₂ impurity after one month exposure and negligible PbI₂ (~7%) from XRD monitoring. However, the degradation beyond 2 months of the pin-hole-free film was quite different from that of the mesoporous as-prepared film without nucleation agent. We found the trace of HPbI₃ after 7 weeks at 2θ=11.6°, and then peaks at 2θ=8.1° and 8.7° indicating (001), (100) facet of

$\text{CH}_3\text{NH}_3\text{PbI}_3 \cdot \text{H}_2\text{O}$ after 9 weeks. This can be well explained by the sequential degradation thermodynamics in combination with degradation kinetics if we consider the surface effect. On the surface of pin-hole-free film, due to the ready release of both HI and CH_3NH_2 , the sequential degradation reactions 2(i), 2(ii), 2(iii) occurred nearly simultaneously and thus only yielded PbI_2 that we observed. When the degradation went to the bulk-like single crystal inside³⁷, CH_3NH_2 and HI were difficult to be released due to high-quality of the film, thus forming HPbI_3 and $\text{CH}_3\text{NH}_3\text{PbI}_3 \cdot \text{H}_2\text{O}$ inside as we detected in XRD patterns after long time degradation. This explanation has been further tested by the following experiments. First, we tried to recover the $\text{CH}_3\text{NH}_3\text{PbI}_3 \cdot \text{H}_2\text{O}$ back to $\text{CH}_3\text{NH}_3\text{PbI}_3$ through heating (Fig.5c). To our surprise, after 12 h heating at 75°C and 2 h heating at 100°C , we found there was not any observable change of monohydrated phase. We increased the calcination temperature gradually and found the change happened at above 110°C which was actually the decomposition threshold of defective perovskite. Second, we also checked the morphology after 70 days degradation and found the degradation products were really embedded in the film (compare Supplementary Fig. 17 with Fig. 4a), which thus confirms the assumption. Finally, we should once again draw the attention to degradation of the perovskite film without nucleation agent above. Due to the mesoporous structure that was fully exposed to moisture, we did not observe the transition products as the bulk-like pin-hole-free film.”

- Fig 1e,f scale bar should be given and how those crystals were prepared.

Answer: The correction has been made with appreciation.

- The authors found that their devices displayed even better performances after being exposed to moisture for 1 month. This finding is very interesting and some comment from the authors on this point would improve the impact of the paper.

Answer: This conscientious comment is adopted with appreciation. The nonstoichiometric acid-base reaction produces high quality films and the slightly doping in the ambient condition could lead to better electronic properties like general doping effect and/or surface passivation in traditional semiconductor and perovskite solar cell. We also quantified the doping concentration in this work.

Please see Page 6 or below:

“The improved performance suggests the thimbleful amount of H_2O benefits degraded perovskite for solar cell efficiency. We have tried to track the doping effect and/or surface passivation on solar performance and quantitatively characterize the concentration of pure perovskite film using external standard method with XRD patterns. (Fig. 6b) The weight concentration was characterized for the

control film and NABR produced film. We can see there are above 90% perovskite remained after about 2 months, and however only ~60% perovskite residue after 7 days degradation along the control method. The best performance of perovskite has doping concentration about 7% by humidity, which is generally consistent with the concentration range in previous work using PbI₂-rich inorganic/organic composition.³⁸

Reviewer #2 (Remarks to the Author):

Please see the attached document.

When reviewing original research please incorporate the points below into your comments to Authors.

For all other article types, such as review or progress articles, please simply provide comments to authors and editors in the boxes provided below.

A. Summary of the key results

B. Originality and interest: if not novel, please give references

C. Data & methodology: validity of approach, quality of data, quality of presentation

D. Appropriate use of statistics and treatment of uncertainties

E. Conclusions: robustness, validity, reliability

F. Suggested improvements: experiments, data for possible revision

G. References: appropriate credit to previous work?

H. Clarity and context: lucidity of abstract/summary, appropriateness of abstract, introduction and conclusions

A. This article presents an alternative synthesis route to make the Organic-Perovskite based photovoltaics that show better moisture resistance. Currently one of the challenges in commercialization of this highly efficient material is its degradation when it is exposed to moisture. An acid based reaction that produces HPbI₃ through excess HI reacting with PbI₂ and subsequent reaction between CH₃NH₂ and HPbI₃ is shown to deliver thin films that are stable in ~65% humidity for up to two months. The results are verified using thin film XRD, TEM, TGA and UV measurements.

B. The work is reasonably original. Some literature search reveals that this method of synthesis has not

been tried before. However, I am not convinced that this is of great general interest as to be published in nature communications.

Answer: We thank referee's comments on the significance. This synthesis method should be of great general interest. See Page 6 or below.

"This new method is beneficial for long-term development and large-scale production under ambient conditions, thus reducing the cost in device fabrication and encapsulation. First, in traditional methods, $\text{CH}_3\text{NH}_3\text{I}+\text{PbI}_2$, the $\text{CH}_3\text{NH}_3\text{I}$ solid is sensitive to humidity and thermal stress, which hinders the up-scaling fabrication under ambient conditions. In the NABR, namely $\text{CH}_3\text{NH}_2+\text{HPbI}_3$, both CH_3NH_2 and HPbI_3 are chemically stable and thus facilitate the fabrication. Moreover, the synthesis of HPbI_3 is much easier than $\text{CH}_3\text{NH}_3\text{I}$ due to the much lower solubility and thus can be collected using ethanol. Second, fundamentally speaking, the reaction between $\text{CH}_3\text{NH}_3\text{I}+\text{PbI}_2$ first yields the intermediate complex of $[\text{PbI}_3]^-$ framework and then sterically arranges the CH_3NH_3^+ . Therefore using well defined HPbI_3 that is stoichiometrically identical to the intermediate complex as starting material is more chemically reasonable. Besides, we have fabricated a large area perovskite solar cell up to $1.0 \times 0.5 \text{ cm}^2$ in area, which successfully delivered 15.0% PCE at present stage, among one of the highest efficiencies for large-area PSC, and thus demonstrated up-scaling potential for solar modules in ambient atmosphere using these solar cells. (Supplementary Fig. 22)"

The other thing to mention here is there is negligible formation of PbI_2 impurity after two months, however what happens after two months? Is this time sufficient for commercialization or is more work required to lengthen this time period. The argument presented is that two months is significantly longer than previous work which showed stability for weeks.

Answer: We thank referee's comments on the stability time and increase some Figures in the paper (see the new Fig. 5, Fig. 6, Fig. S17). Actually, the two-month stability is sufficient for the following fabrication into solar cell. Besides, ~65% humidity in the test is generally higher than other reports. According to our control experiment, under 40% humidity test, it is stable for even longer than 3 months. Due to much lower than ambient humidity, it's beyond our discussion in this work. Besides, if with slight protection, such as simple encapsulation in solar cell, the stability is even better according to previous reports (Ref. 20, 21). Therefore, our intrinsically stable perovskite film will greatly reduce the encapsulation cost in devices. We also discussed the different degradation process happened beyond two months, which

helped to understand the stability properties of perovskite more clearly.

“We have found that optimized perovskite thin film with nucleation agent remained stable through color and XRD monitoring. (Fig. 5b) Under ~65% moisture, there was no signature of PbI_2 impurity after one month exposure and negligible PbI_2 (~7%) from XRD monitoring. However, the degradation beyond 2 months of the pin-hole-free film was quite different from that of the mesoporous as-prepared film without nucleation agent. We found the trace of HPbI_3 after 7 weeks at $2\theta=11.6^\circ$, and then peaks at $2\theta=8.1^\circ$ and 8.7° indicating (001), (100) facet of $\text{CH}_3\text{NH}_3\text{PbI}_3\cdot\text{H}_2\text{O}$ after 9 weeks. This can be well explained by the sequential degradation thermodynamics in combination with degradation kinetics if we consider the surface effect. On the surface of pin-hole-free film, due to the ready release of both HI and CH_3NH_2 , the sequential degradation reactions 2(i), 2(ii), 2(iii) occurred nearly simultaneously and thus only yielded PbI_2 that we observed. When the degradation went to the bulk-like single crystal inside³⁷, CH_3NH_2 and HI were difficult to be released due to high-quality of the film, thus forming HPbI_3 and $\text{CH}_3\text{NH}_3\text{PbI}_3\cdot\text{H}_2\text{O}$ inside as we detected in XRD patterns after long time degradation. This explanation has been further tested by the following experiments. First, we tried to recover the $\text{CH}_3\text{NH}_3\text{PbI}_3\cdot\text{H}_2\text{O}$ back to $\text{CH}_3\text{NH}_3\text{PbI}_3$ through heating (Fig.5c). To our surprise, after 12 h heating at 75°C and 2 h heating at 100°C , we found there was not any observable change of monohydrated phase. We increased the calcination temperature gradually and found the change happened at above 110°C which was actually the decomposition threshold of defective perovskite. Second, we also checked the morphology after 70 days degradation and found the degradation products were really embedded in the film (compare Supplementary Fig. 17 with Fig. 4a), which thus confirms the assumption. Finally, we should once again draw the attention to degradation of the perovskite film without nucleation agent above. Due to the mesoporous structure that was fully exposed to moisture, we did not observe the transition products as the bulk-like pin-hole-free film.”

There are other work referenced here which can accomplish similar stability by cell design [ref 20]. Why is that not the correct approach for the fabrication of cells? Would this process suggest here be more cost effective?

Answer: The stability in cell is also convincing in Ref. 20. In this work, we have demonstrated intrinsic stability because it is more challenging at present than device stability. The cell encapsulation and shielding layer to resist moisture are facile to improve stability. However, if the material stability can't be high, the encapsulation would be strictly performed with high cost. The improved material stability will benefit for the following fabrication of devices and we demonstrate it through chemistry reaction control and engineering. Compared to Ref. 20, the much longer moisture resistance time (60 vs 6 days) and higher humidity degree (65% vs 55%) for perovskite material are two basic improvements.

See page 6 or below:

“Besides, it has to mention that stable perovskite solar cell through shielding layer in previous work is generally lower than 16.2%.^{20,39} According to their detailed reports, perovskite itself did not endure high moisture (>55%) for 1-6 days without encapsulation of these high efficiency perovskite photovoltaics, which indicates that the high stability primarily comes from encapsulation. However, here we demonstrate that NABR using excess CH_3NH_2 reaction with well-defined HPbI_3 is a reliable route to produce high material-stable perovskite than previous work.^{20,39}”

Is the synthesis scalable to large production? Given the study is motivated by large scale production, some of these discussion should be included to strengthen the novelty and application.

Answer: The practical advantages and scale-up potential of our synthesis methods have been clarified with many thanks for the comment. See Page 6 or below.

“This new method is beneficial for long-term development and large-scale production under ambient conditions, thus reducing the cost in device fabrication and encapsulation. First, in traditional methods, $\text{CH}_3\text{NH}_3\text{I}+\text{PbI}_2$, the $\text{CH}_3\text{NH}_3\text{I}$ solid is sensitive to humidity and thermal stress, which hinders the up-scaling fabrication under ambient conditions. In the NABR, namely $\text{CH}_3\text{NH}_2+\text{HPbI}_3$, both CH_3NH_2 and HPbI_3 are chemically stable and thus facilitate the fabrication. Moreover, the synthesis of HPbI_3 is much easier than between $\text{CH}_3\text{NH}_3\text{I}+\text{PbI}_2$ first yields the intermediate complex of $[\text{PbI}_3]^-$ framework and then sterically arranges the CH_3NH_3^+ . Therefore using well defined HPbI_3 that is stoichiometrically identical to the intermediate complex as starting material is more chemically reasonable. Besides, we have fabricated a large area perovskite solar cell up to $1.0 \times 0.5 \text{ cm}^2$ in area, which successfully delivered 15.0% PCE at present stage, among one of the highest efficiencies for large-area PSC, and thus demonstrated up-scaling potential for solar modules in ambient atmosphere using these solar cells. (Supplementary Fig. 22)”

C. Most of the data is used in more of a qualitative than quantitative analysis. Often phase identification is done on the basis of one reflection. Since the work done is on thin film, the crystallographic models can only be guessed but a full model refinement on bulk would be more convincing for the synthesis mechanism suggested. There is no mention of the film thickness which can be important for application. So most of what is being discussed is surface effects. However to confirm the hypothesis in a stronger platform would need looking at the bulk as well.

Answer: Referring to quantitative analysis, we would like to add the quantitative comparison between most stable perovskite film with our method and the traditional product with one-step method, from which we can see the much more stable product along our new method. Please

see Page 6 or below:

"The improved performance suggests the thimbleful amount of H₂O benefits degraded perovskite for solar cell efficiency. We have tried to track the doping effect and/or surface passivation on solar performance and quantitatively characterize the concentration of pure perovskite film using external standard method with XRD patterns. (Fig. 6b) The weight concentration was characterized for the control film and NABR produced film. We can see there are above 90% perovskite remained after about 2 months, and however only ~60% perovskite residue after 7 days degradation along the control method. The best performance of perovskite has doping concentration about 7% by humidity, which is generally consistent with the concentration range in previous work using PbI₂-rich inorganic/organic composition.³⁸

Second, as we develop synthetic route for perovskite photovoltaic film, the bulk property is not applicable for films. We would like to investigate the bulk properties in the following work to facilitate deep and complete research insight. In present work, the thickness thin film is generally 350-450 nm for photovoltaics.

The presentation of the material is reasonable, however, a large number of acronyms have been used without first spelling them out (i.e. DMSO, PL, PSC).

Answer: The correction has been made with appreciation.

D. As mentioned earlier, very little data fitting has been applied and most of the data is used in a qualitative manner so not much statistical treatment is presented in the paper.

Answer: The correction has been made with appreciation referring to previous comment.

E. Conclusions: Conclusions are supported by the observation presented.

Answer: We thank the reviewer for the comment.

F. Page two, the sentence "Based on previous work by others....." needs references

Answer: The correction has been made with appreciation.

Equation (2-i) does not make sense. How can $A + B = B - A$? Also given the crux of the paper is a new synthesis mechanism, clearly one would like to know what is the final yield of the target product?

Answer: We are sorry about this point and have revised them with many thanks. Actually, we use “-” as chemical bond but not minus. We changed “-” to “•” for clarification with appreciation. See page 2 or below.

“**Recoverable Degradation.** In order to find the way to improve the stability of $\text{CH}_3\text{NH}_3\text{PbI}_3$, we have first investigated the degradation of traditional perovskites and employed the recent defect-healing process for the recovery of degraded perovskite to check the transition products.²⁷ Figs. 1a and 1b showed the degraded and recovered perovskite thin film (prepared by traditional two-step method). We found that after degradation for some time, perovskite became yellow (Fig.1b) and had the distinct XRD pattern of PbI_2 at $2\theta=12.6^\circ$. Although we thought it was fully degraded, it was immediately recovered by CH_3NH_2 vapor. The CH_3NH_2 -recovered perovskite film was confirmed by the XRD peaks (Fig. 1a) at $2\theta=14.1^\circ$, 28.4° that were indicative of $\text{CH}_3\text{NH}_3\text{PbI}_3$ (110), (220). The recovered perovskite films existed in the form of nanoscale crystals judging from the broad XRD peaks. The photoluminescence (PL) mapping contrasts for the degraded (left) and recovered (right) films indicated the uniform reversion to perovskite (right). In comparison, the degraded film prepared by traditional one-step was employed for recovery test. These films had fast degradation rates and yielded some transition product under observation during the degradation. Figs. 1c and 1d show the basic results. After 3-day degradation in 65% humidity, we observed a small angle XRD peak at $2\theta=8.1^\circ$, which was identical to the hydrated-HPbI₃ complex ($\text{CH}_3\text{NH}_3\text{PbI}_3 \cdot \text{H}_2\text{O}$, with XRD peaks at $2\theta=8.10^\circ$, 8.66° , and 10.66°).^{23,25,28} After 3 weeks, the degraded film had only PbI_2 peak at $2\theta=12.6^\circ$. However, these films were also recovered to perovskite nanoscale crystals, (Fig.1c) with PL mapping so as to confirm the uniform recovery judged from the strong PL at 760 nm. (Fig.1d)²⁹ The similar process of degradation and recovery was also observed in the mixed “ $\text{CH}_3\text{NH}_3\text{PbI}_{3-x}\text{Cl}_x$ ” perovskite using 3:1 mole combination of $\text{CH}_3\text{NH}_3\text{I}$ and PbCl_2 , which was high-performing in PSC but presumably the most unstable perovskite compared to the iodide perovskite. We saw that the degradation process clearly exhibited the transition product of hydrated-HPbI_{3-x}Cl_x ($\text{CH}_3\text{NH}_3\text{PbI}_{3-x}\text{Cl}_x \cdot \text{H}_2\text{O}$) in the degradation, as shown in small angle of XRD at $2\theta=8.10^\circ$, 8.66° , and 10.66° , corresponding to the (001), (100), and ($\bar{1}01$) reflections of a monoclinic P2₁/m crystal structure, and could be recovered to some extent. (Supplementary Fig. 1)

This general recovery by methylamine suggested the degraded film with PbI_2 and solvated-HPbI₃ could be reconverted to perovskite using methylamine and could infer that degradation of perovskite was due to the loss and/or lack of methylamine. In order to evaluate the reaction process of perovskite more clearly, the large crystals were prepared along two-step method for *in-situ* observation of the recovery via optical microscope. We found that after degradation, the yellow phase crystals did not have too much different morphology from that of the parent black perovskite. (Supplementary Fig. 2) After recovery using CH_3NH_2 , the degraded crystals re-crystallized into much smaller and more compact grains. (Fig. 1e, 1f)

In order to illustrate more clearly, we approximately analysed the chemical reactions for perovskite formation with dimethylformamide (DMF) solution and degradation in the following.

In the formation:

Based on previous work,^{25,28} we observed the colloidal characteristics and redshift of perovskite precursor compared to PbI₂, which verified (1-i) or (1-ii) were right. The products of both (1-i) and (1-ii) could yield (1-iii). The CH₃NH₃PbI₃·DMF has been detected by XRD in other report²⁸ and will be confirmed in the following. The DMF could be replaced by isopropanol (IPA) along the traditional two-step route. Noted that these routes could yield the possible byproducts below due to the coordination of DMF and removal afterwards:

In the degradation, we could simply write the following reactions:

Actually, (2-i) has been verified through *in-situ* time resolved XRD techniques and heating-recovery in previous work.³⁰ Reaction (2-iii) can be easily concluded from the final products. Degradation reaction (2-ii) was apparently judged from CH₃NH₂-recovery in the glove box (Supplementary Fig. 3) and directly proved in the following. Therefore, we conclude that in the humidity degradation, the perovskite is sequentially decomposed in terms of thermodynamics, first to an intermediate hydrated-HPbI₃ complex (2-i), and then to release fade-away CH₃NH₂ molecules (2-ii), finally to PbI₂ solid and HI/H₂O vapor (2-iii), although the different kinetics could produce different degraded compounds in final products (such as CH₃NH₃PbI₃·H₂O, PbI₂ and HPbI₃ etc).^{23,30}

Throughout the document the small angle has been wrongly used. When one talks about small angle in x-ray, 2 theta. In the scattering community small angle has a very different meaning. It is recommended that this is presented in natural units for structural measurement which is generally d-spacing.

Answer: We thank you for the taking notice of small/low angle XRD. Actually, we don't employ it to characterize the d-spacing in mesoscopic structure. All 2theta values generally lie between 5 and 70 degree, thus we still use the 2theta for facilitating the comparison with previous work in this area. (such as Ref 23, 28, Ref 23 "characterized by low angle X-ray diffraction reflections (typically 2θ < 10°)," Ref 28" Several new peaks arise at low angle that cannot be assigned to either CH₃NH₃PbI₃ or PbI₂. Specifically, new peaks are seen at 7.93°, 8.42°, 10.46°, and 16.01°) "

G. See F. The references are appropriate and give credit to previous work.

Answer: The correction has been made with appreciation.

H. As mentioned earlier, the introduction needs to discuss about scale up and cost effectiveness, given the main argument here is this is needed for commercialization of solar cells made with these materials.

Answer: The practical advantages and scale-up potential of our synthesis methods have been clarified with appreciation for the comment. We changed some of the introduction and also added a paragraph to response this comment. See Page 2 and 6 or below.

Page 2. " ...This work provides the important insight into perovskite intrinsic stability and the utilization of simple chemical reaction for material control with scaling-up potential in PSC."

Page 6. "This new method is beneficial for long-term development and large-scale production under ambient conditions, thus reducing the cost in device fabrication and encapsulation. First, in traditional methods, $\text{CH}_3\text{NH}_3\text{I}+\text{PbI}_2$, the $\text{CH}_3\text{NH}_3\text{I}$ solid is sensitive to humidity and thermal stress, which hinders the up-scaling fabrication under ambient conditions. In the NABR, namely $\text{CH}_3\text{NH}_2+\text{HPbI}_3$, both CH_3NH_2 and HPbI_3 are chemically stable and thus facilitate the fabrication. Moreover, the synthesis of HPbI_3 is much easier than $\text{CH}_3\text{NH}_3\text{I}$ due to the much lower solubility and thus can be collected using ethanol. Second, fundamentally speaking, the reaction between $\text{CH}_3\text{NH}_3\text{I}+\text{PbI}_2$ first yields the intermediate complex of $[\text{PbI}_3]^-$ framework and then sterically arranges the CH_3NH_3^+ . Therefore using well defined HPbI_3 that is stoichiometrically identical to the intermediate complex as starting material is more chemically reasonable. Besides, we have fabricated a large area perovskite solar cell up to $1.0 \times 0.5 \text{ cm}^2$ in area, which successfully delivered 15.0% PCE at present stage, among one of the highest efficiencies for large-area PSC, and thus demonstrated up-scaling potential for solar modules in ambient atmosphere using these solar cells. (Supplementary Fig. 22)"

Reviewer #3 (Remarks to the Author):

In the presented paper, degradation of the methylammonium lead iodide is briefly analyzed to propose the facile two-step non-stoichiometric acid-base route of its thin film synthesis. Colloidal stability of precursors (HPbI₃, MA+HPbI₃, and control MAI+PbI₂) studied as well as the stability of materials in time and under humidity. Authors propose HPbI₃ as an intermediate compound. The significance of the MAPbI₃ thin film formation study is important due to the rising issue of a long-term stability of PV devices based on perovskite materials. Authors mainly are trying to solve the problem of volume-expansion control during the two-step synthesis and the obtained perovskite is appeared to resist degradation judging from XRD patterns measured after 56 days. The paper is novel and interesting because it brings to light the role of HI excess during the first step and the influence of the synthetic route on the morphology of the thin film.

Answer: We thank referee's comments and we will improve the negative aspects of the paper with point-to-point response in the following.

Still a few inconsistencies can be noticed. First, authors quickly jump to the conclusion that HPbI₃ is the intermediate step during the synthesis, whereas many authors have been more cautious to make such prediction. Though there have been evidences of the enhanced coordination in the HI + PbI₂ + DMF solution in the presented paper (colloidal behavior of the latter solution; red-shift in the absorbance spectrum; expansion of the lattice spacing comparing to PbI₂ in the deposited thin film), the exact mechanism still remains unclear. It may be that the role of the solvent DMF and its intercalation between PbI₂ sheets is underestimated here. Nevertheless, there is no doubt that the I-coordination occurs in HI-rich solutions of PbI₂ and the paper opens the prospect for

the further investigations (as well as investigation of the structure of these intermediate complex and the role of proton). In addition, recently HPbI₃ was also mentioned elsewhere[1], denoting that the role of polyplumbate complexes as an intermediate step in the synthesis of perovskites might be of a certain interest.

Answer: We thank the referee's kind suggestion on the consideration of role of DMF and clarifying the role of HPbI₃. HPbI₃ is indeed not the direct intermediate complex but has stoichiometrically identical PbI₃ to the intermediate solvated-HPbI₃ complex and perovskite. HPbI₃ has full coordination and adopts a monoclinic structure employing [PbI₃]⁻ double chain with edge sharing. The DMF will help to dissolve the HPbI₃ through intercalation in the solution and form solvate during the spin-coating. Therefore, we changed the expression for accuracy with appreciation.

Page 1. "Besides, first-step produced HPbI₃ was stoichiometrically identical to intermediate solvated-HPbI₃ complex in traditional method (CH₃NH₃I+PbI₂) and afforded to promote the second-step reaction."

Page 3. "In reaction (3), the excess hydroiodic acid promotes the reaction completely and yields HPbI₃ that is stoichiometrically identical to [PbI₃]⁻ framework of perovskite, permitting to form fully coordinated framework for perovskite without I-vacancy. In reaction (4), the excess CH₃NH₂ allows the full reconversion of HPbI₃ to perovskite and thus eliminated the CH₃NH₂ vacancy. Different from use of excess CH₃NH₃I reacting with PbI₂, both CH₃NH₂ and HI are facile to remove towards stoichiometric lattice due to the volatility at room temperature. Besides, the two reactions are probably exothermic due to acid-base neutralization and could occur spontaneously."

Page 3,4 " We have then correlated the morphology with crystallographic information in detail through XRD monitoring (Fig. 3c and 3d). The wet PbI₂ film had small angle peak at 2θ=9.5°, which was due to the coordination of DMF (PbI₂·DMF) (Fig. 3c black curve) that was similar to the coordination of PbI₂ with dimethyl sulfoxide (DMSO) that was PbI₂(DMSO).⁹ This soft coordination complex facilitated the film formation of PbI₂ as aforementioned through gradually releasing DMF for slow crystallization during spin-coating without baking (Fig. 3c green curve). The baking increased the crystallinity from the strong (001) peak at 2θ=12.6° and observable (003) peak at 2θ=38.6° (Fig. 3c red curve). HPbI₃ crystallized quickly in the wet film and displayed the same XRD pattern at 2θ=11.6° to the final products, which meant

the DMF was not strongly bonded to HPbI₃ due to HI coordination but just intercalated with weak interaction. This fast crystallization was however detrimental for uniform film formation, resulting in poor coverage and rough surface of thin film as afore-discussed. The HPbI₃ had increased the lattice spacing of 7.58 Å referring to 2θ=11.6° compared to interlayer spacing of 6.98 Å along the c-axis (001) of PbI₂. The transition products of perovskites were recognized by the small angle (low angle) XRD peaks in wet films at 2θ=6.5°, 7.9° and 9.4°, corresponding to solvated-HPbI₃ (CH₃NH₃PbI₃·xDMF and/or CH₃NH₃PbI₃·xCH₃NH₂, etc.). However, in the spin-coated films before baking, these small angle peaks of NABR film disappeared, but the control sample of one-step still had such peaks if without baking, which meant that DMF was still incorporated in the lattice for the one-step sample. This information suggests that there exists iodine/CH₃NH₂ deficiency in the perovskite lattice in the control sample and thus DMF is not easy to be removed at room temperature. NABR method reduces the vacancies owing to full coordination of HI and CH₃NH₂ solvation (CH₃NH₃PbI₃·x(CH₃NH₂)). Namely, DMF molecules just play the role in solvation and/or intercalation but not direct coordination with lead ions in NABR.”

Second, the degradation mechanism presented in the paper is highly simplified paying attention to the fact that the recent crystallographic study [2] revealed formation of monohydrate and more complex dihydrate during the decomposition under humidity. The oversimplification may be the main issue leading to misinterpretations.

Answer: This comment points out the more complicated things that really exist in the degradation, which is highly related to thermodynamics (Energy states) and kinetics (Reaction rates and pathways). In Ref. 23 [2] and 30, these authors believed the monohydrate and dihydrate were two basic degradation products. In Ref. 23 [2], the authors stated that “H₂O is able to complex with the perovskite, forming a hydrate product similar to (CH₃NH₃)₄PbI₆·2H₂O (Ref 23)”. In Ref. 30, the authors assigned 2θ=11.6 degree for degradation compound due to the similar XRD

position of $(\text{CH}_3\text{NH}_3)_4\text{PbI}_6 \cdot 2\text{H}_2\text{O}$ at $2\theta = 11.4$ degree. Therefore, the reviewer reminded us to pay attention to the fact that the recent crystallographic study [2] revealed formation of monohydrate and more complex dihydrate during the decomposition under humidity, aiming to making the degradation more clear and reducing the misinterpretation. Through careful comparison their XRD results, we think that the monohydrate is convincing since our results also fully support it (see the revisions in page 3 or below). However, the second degraded compound is more likely to be HPbI_3 rather than **$(\text{CH}_3\text{NH}_3)_4\text{PbI}_6 \cdot 2\text{H}_2\text{O}$** based on the following points. First, HPbI_3 has identical XRD peak at $2\theta = 11.6$ degree to the mentioned second degradation product (This work in the Fig 5 and Ref. 30). Second, due to the extraction of MA by moisture for solubility equilibrium, there is no additional MA to produce **$(\text{CH}_3\text{NH}_3)_4\text{PbI}_6 \cdot 2\text{H}_2\text{O}$ from reaction analysis**. Third, if we think it's really $(\text{CH}_3\text{NH}_3)_4\text{PbI}_6 \cdot 2\text{H}_2\text{O}$, $(\text{CH}_3\text{NH}_3)_4\text{PbI}_6 \cdot 2\text{H}_2\text{O}$ is not stable and will further degrade to final PbI_2 due to the loss of MA, HI and H_2O , which also implied solvate HPbI_3 is an intermediate product. The fourth, we should draw attention to the fact that MA has much high solubility (12 M) than HI (7.8 M), which means the extraction of MA by moisture is preferable to HI, thus the degradation should sequentially produce monohydrate $\text{MAPbI}_3 \cdot \text{H}_2\text{O}$, then HPbI_3 and finally PbI_2 . Moreover, through the dipping the perovskite film into water, or in condensed moisture (80%, in Ref. 30), it does not produce dihydrated $(\text{CH}_3\text{NH}_3)_4\text{PbI}_6 \cdot 2\text{H}_2\text{O}$ but produce PbI_2 , which also indicates the loss of MA into moisture and then HI. Therefore, through the direct XRD assignment and further discussion for Fig. 5, we believe degradation mechanism should be much more reasonable right now. We greatly thank this comment for addressing degradation mechanism of perovskite with reduced misinterpretation, which will attract broad attention.

Page 3. “Recoverable Degradation. In order to find the way to improve the stability of $\text{CH}_3\text{NH}_3\text{PbI}_3$, we have first investigated the degradation of traditional perovskites and employed the recent defect-healing process for the recovery of degraded perovskite to check the transition products.²⁷ Figs. 1a and 1b showed the degraded and recovered perovskite thin film (prepared by traditional two-step method). We found that after degradation for some time, perovskite became yellow (Fig.1b) and had the distinct XRD pattern of PbI_2 at $2\theta=12.6^\circ$. Although we thought it was fully degraded, it was immediately recovered by CH_3NH_2 vapor. The CH_3NH_2 -recovered perovskite film was confirmed by the XRD peaks (Fig.1a) at $2\theta=14.1^\circ$, 28.4° that were indicative of $\text{CH}_3\text{NH}_3\text{PbI}_3$ (110), (220). The recovered perovskite films existed in the form of nanoscale crystals judging from the broad XRD peaks. The photoluminescence (PL) mapping contrasts for the degraded (left) and recovered (right) films indicated the uniform reversion to perovskite (right). In comparison, the degraded film prepared by traditional one-step was employed for recovery test. These films had fast degradation rates and yielded some transition product under observation during the degradation. Figs. 1c and 1d show the basic results. After 3-day degradation in 65% humidity, we observed a small angle XRD peak at $2\theta=8.1^\circ$, which was identical to the hydrated-HPbI₃ complex ($\text{CH}_3\text{NH}_3\text{PbI}_3 \cdot \text{H}_2\text{O}$, with XRD peaks at $2\theta=8.10^\circ$, 8.66° , and 10.66°).^{23,25,28} After 3 weeks, the degraded film had only PbI_2 peak at $2\theta=12.6^\circ$. However, these films were also recovered to perovskite nanoscale crystals, (Fig.1c) with PL mapping so as to confirm the uniform recovery judged from the strong PL at 760 nm. (Fig.1d)²⁹ The similar process of degradation and recovery was also observed in the mixed “ $\text{CH}_3\text{NH}_3\text{PbI}_{3-x}\text{Cl}_x$ ” perovskite using 3:1 mole combination of $\text{CH}_3\text{NH}_3\text{I}$ and PbCl_2 , which was high-performing in PSC but presumably the most unstable perovskite compared to the iodide perovskite. We saw that the degradation process clearly exhibited the transition product of hydrated-HPbI_{3-x}Cl_x ($\text{CH}_3\text{NH}_3\text{PbI}_{3-x}\text{Cl}_x \cdot \text{H}_2\text{O}$) in the degradation, as shown in small angle of XRD at $2\theta=8.10^\circ$, 8.66° , and 10.66° , corresponding to the (001), (100), and ($\bar{1}01$) reflections of a monoclinic P2₁/m crystal structure, and could be recovered to some extent. (Supplementary Fig. 1)

This general recovery by methylamine suggested the degraded film with PbI_2 and solvated-HPbI₃ could be reconverted to perovskite using methylamine and could infer that degradation of perovskite was due to the loss and/or lack of methylamine. In order to evaluate the reaction process of perovskite more clearly, the large crystals were prepared along two-step method for *in-situ* observation of the recovery via optical microscope. We found that after degradation, the yellow phase crystals did not have too much different morphology from that of the parent black perovskite. (Supplementary Fig. 2) After recovery using CH_3NH_2 , the degraded crystals re-crystallized into much smaller and more compact grains. (Fig. 1e, 1f)

In order to illustrate more clearly, we approximately analysed the chemical reactions for perovskite formation with dimethylformamide (DMF) solution and degradation in the following.

In the formation:

Based on previous work,^{25,28} we observed the colloidal characteristics and redshift of perovskite precursor compared to PbI_2 , which verified (1-i) or (1-ii) were right. The products of both (1-i) and (1-ii) could yield (1-iii). The $\text{CH}_3\text{NH}_3\text{PbI}_3 \cdot \text{DMF}$ has been detected by XRD in other report²⁸ and will be confirmed in the following. The DMF could be replaced by isopropanol (IPA) along the traditional two-step route. Noted that these routes could yield the possible byproducts below due to the coordination of DMF and removal afterwards:

In the degradation, we could simply write the following reactions:

Actually, (2-i) has been verified through *in-situ* time resolved XRD techniques and heating-recovery in previous work.³⁰ Reaction (2-iii) can be easily concluded from the final products. Degradation reaction (2-ii) was apparently judged from CH₃NH₂-recovery in the glove box (Supplementary Fig. 3) and directly proved in the following. Therefore, we conclude that in the humidity degradation, the perovskite is sequentially decomposed in terms of thermodynamics, first to an intermediate hydrated-HPbI₃ complex (2-i), and then to release fade-away CH₃NH₂ molecules (2-ii), finally to PbI₂ solid and HI/H₂O vapor (2-iii), although the different kinetics could produce different degraded compounds in final products (such as CH₃NH₃PbI₃·H₂O, PbI₂ and HPbI₃ etc).^{23, 30}

Page 5: " We have further performed TGA to check thermal stability of perovskite films by different methods, as well as starting materials: HPbI₃, CH₃NH₃I and PbI₂. TGA curve for CH₃NH₃I shows nearly 100% weight loss between 260 °C and 320 °C for CH₃NH₃I, respectively. (Supplementary Fig. 12a) HPbI₃ has a large weight loss at low temperature between 300 °C and 360 °C indicative of release of HI. Consistent with humidity stability, the perovskite prepared by one-step method was also not thermally stable. The weight loss onset was at 60 °C, which was consistent with previous report²² and meant organic-inorganic components of the one-step prepared perovskite were not tightly bonded. Through careful observation, the sequential thermal decomposition could be identified (see reaction 5) judged from two different weight loss regions between 60 °C-150 °C and 250 °C-350 °C. For the NABR, the weight loss of organic component was much larger than the others, indicative of fully coordinated [PbI₃]⁻ scaffold and enough CH₃NH₂ filling in the lattice.

Probably, perovskite prepared by the two-step method and NABR also decomposed sequentially under heating stress in terms of thermodynamics. Due to the similar release rates of MA and HI, we were unable to detect sequential events in terms of kinetics. In principle, the sequential decomposition thermodynamics is acceptable because when iodine well bonded to PbI₂, the bond energy of MA-I is reduced and could be regarded as hydrogen bond. In the one-step prepared film, large number of vacancies accelerates the decomposition rates of MA, allowing us to observe the sequential loss.

TGA was performed in low temperature region for the release of the organic component, with 30 min heat preservation at 100 °C and 120 min heat preservation at 200 °C for the observation of weight loss clearly. We can see the weakest thermal stability of perovskite prepared by one-step, robust thermal stability for fully converted perovskite by two-step and tunable stability through CH₃NH₂ composition control in the NABR. (Supplementary Fig. 12b) Besides, the NABR affords to endure high temperature calcination under below 150 °C for high quality films. (Supplementary Fig. 13) Therefore, the NABR represents a controllable way towards highly stable perovskite in the humidity and under heat stress. "

Page 5

"We have found that perovskite remained stable through color and XRD monitoring. (Fig. 5b) Under ~65% moisture, there was no signature of PbI₂ impurity after one month exposure and negligible PbI₂ (~7%) from XRD monitoring. However, the degradation beyond 2 months of the pin-hole-free film was quite different from that of the mesoporous as-prepared film without nucleation agent. We found the trace of HPbI₃ after 7 weeks at 2θ=11.6°, and then peaks at 2θ=8.1° and 8.7° indicating (001), (100) facet of CH₃NH₃PbI₃·H₂O after 9 weeks. This can be well explained by the sequential degradation thermodynamics in combination with degradation kinetics if we consider the surface effect. On the surface of pin-hole-free film, due to the ready release of both HI and CH₃NH₂, the sequential degradation reactions 2(i), 2(ii), 2(iii) occurred nearly simultaneously and thus only yielded PbI₂ that we observed. When the degradation went to the bulk-like single crystal inside³⁷, CH₃NH₂ and HI were difficult to be released due to high-quality of the film, thus forming HPbI₃ and CH₃NH₃PbI₃·H₂O inside as we detected in XRD patterns after long time degradation.

This explanation has been further tested by the following experiments. First, we tried to recover the $\text{CH}_3\text{NH}_3\text{PbI}_3 \cdot \text{H}_2\text{O}$ back to $\text{CH}_3\text{NH}_3\text{PbI}_3$ through heating (Fig.5c). To our surprise, after 12 h heating at 75°C and 2 h heating at 100°C , we found there was not any observable change of monohydrated phase. We increased the calcination temperature gradually and found the change happened at above 110°C which was actually the decomposition threshold of defective perovskite. Second, we also checked the morphology after 70 days degradation and found the degradation products were really embedded in the film (compare Supplementary Fig. 17 with Fig. 4a), which thus confirms the assumption. Finally, we should once again draw the attention to degradation of the perovskite film without nucleation agent above. Due to the mesoporous structure that was fully exposed to moisture, we did not observe the transition products as the bulk-like pin-hole-free film.”

Moreover, each of the two-step reactions of the acid-base precursor preparation is proposed to be exothermic without no significant evidences or references.

Answer: Thanks for the comment and we have changed the expression. “Besides, the two reactions were

probably exothermal due to acid-base neutralization and could occur spontaneously.”

Actually, we found the reaction of MA

(g)+ HPbI_3 (s) occur spontaneously. When we transfer the HPbI_3 films into MA vapor and take them out, HPbI_3 immediately turns

dark red indicative of perovskite in the room temperature. For the degradation, the moisture exposure always takes much longer time,

which suggests MA has larger bind energy to HPbI_3 than H_2O . Besides, for the two sequential reactions, we have performed

systematic DFT calculation and will present the exothermic properties in another work with detailed discussion. (see below for

reference) (MA, MAI, HI are treated as gas and the others are solid, energy difference unit: eV)

In addition, few technical flaws have to be mentioned. Among them are sporadically used incorrect phrasal constructions (e.g. "...until we saw the precipitation in the naked eyes..."), which do not facilitate the quick understanding of the text. Also, the element "Figure 1 (e,f)" depicting degraded perovskite crystals does not have a scale to compare the real size. The same remark applies to a few figures in the Supplementary (S7, S19).

Answer: These flaws have been improved with appreciation. "...until we saw the precipitation in the naked eyes..." has been changed to "until it precipitated". We have added the scale bar in the figure.

Besides small flaws and too generalized conclusions, technical side of the paper remains on the high level and the topic of the paper is important for the investigation of the perovskite stability in terms of a large-scale manufacturing. Authors managed to show a remarkable PV performance of the constructed devices and the stability of the perovskite material.

Answer: We thank referee again for comments on the positive side to affirm our work as well as pointing out these flaws for revision, which help to enhance the overall quality of the paper.

[1] Christians, Jeffrey A., Pierre A. Miranda Herrera, and Prashant V. Kamat. "Transformation of the excited state and photovoltaic efficiency of CH₃NH₃PbI₃

perovskite upon controlled exposure to humidified air." *Journal of the American Chemical Society* 137.4 (2015): 1530-1538.

[2] Wang, Feng, et al. "HPbI3: A New Precursor Compound for Highly Efficient Solution-Processed Perovskite Solar Cells." *Advanced Functional Materials* 25.7 (2015): 1120-1126.

Reviewer #1 (Remarks to the Author)

The authors have responded adequately to most of the reviewer comments. The manuscript contains important and novel findings for the perovskite community, hence it should be suitable for Nature Communications. However, the revised manuscript is riddled with improper linguistic and grammatical constructs, therefore it should be proofread and edited carefully by the authors before it could be suitable for publication.

Reviewer #2 (Remarks to the Author)

This article is a resubmission with corrections. The authors have satisfactorily answered my objections and the article is now significantly improved. I recommend publication.

Reviewer #3 (Remarks to the Author)

I had now read a revised manuscript. I appreciate the response to my initial queries. Still, important revisions need to be done before the paper can be considered for acceptance.

In the first place, I would strongly recommend doing additional experiments on structure determination of the intermediate precursor if it is possible to obtain the bulk material from PbI₂-HI the solution in DMF. Authors mentioned in their response letter that "HPbI₃ has full coordination and adopts a monoclinic structure employing [PbI₃] double chain with edge sharing", but this is not evidenced by single-crystal X-ray diffraction experiments. Moreover, in the previously mentioned work by Wang et al. (2015) XRD data from HPbI₃ is also present and authors may comment on the similarities or differences.

Secondly, authors might discuss the work of Li et al. (2015) as it is connected to their work. Also, some minor inconsistencies in the text are confounding. For example, the phrase "hydrated-HPbI₃ (CH₃NH₃PbI₃·H₂O)" doesn't make much sense. Moreover, in the Discussion paragraph, one can see the following: "...under heating stress, releasing fade-away CH₃NH₂ molecule, then PbI₂ and HI vapor". At which temperature were the degradation experiments conducted so to produce the PbI₂ vapor?

The paper is lucid in general terms, however, some minor and major queries, mentioned above, are aimed to improve the credibility of the paper.

To conclude, the reason for publishing the paper is to report about outstanding long-lasting performance in the presence of humidity and new interesting way of MAPbI₃ synthesis. This should probably be the keynote of the paper and compared more to other methods of protracting the working time of the MAPbI₃ thin-films. On the other hand, as it has been already mentioned in the first revision, the conclusions made on the structure of the intermediate compound, HPbI₃ are brusque and based only on XRD and UV-absorptions measurements, which suggest that the results can't be fully interpreted. The previous revision didn't improve the paper to the level when it should be published in the journal with such high credibility. On the other hand, one can't deny the results are curious and interesting, though they are in the nascent form, and should be definitely published if not in Nature Communications then elsewhere.

Reviewer #1 (Remarks to the Author):

The authors have responded adequately to most of the reviewer comments. The manuscripts contains important and novel findings for the perovskite community, hence it should be suitable for Nature Communications. However, the revised manuscript is riddled with improper linguistic and grammatical constructs, therefore it should be proofread and edited carefully by the authors before it could be suitable for publication.

Reply: We thank the Reviewer for the comment. We have thoroughly revised our manuscript to remove errors in English usage, which addressed improper linguistic and grammatical expression.

Reviewer #2 (Remarks to the Author):

This article is a resubmission with corrections. The authors have satisfactorily answered my objections and the article is now significantly improved. I recommend publication.

Reply: We thank the Reviewer for the comment.

Reviewer #3 (Remarks to the Author):

Point 1:

I had now read a revised manuscript. I appreciate the response to my initial queries. Still, important revisions need to be done before the paper can be considered for acceptance.

In the first place, I would strongly recommend doing additional experiments on structure determination of the intermediate precursor if it is possible to obtain the bulk material from PbI_2 -HI the solution in DMF. Authors mentioned in their response letter that "HPbI₃ has full coordination and adopts a monoclinic structure employing [PbI₃] double chain with edge sharing", but this is not evidenced by single-crystal X-ray diffraction experiments. Moreover, in the previously mentioned work by Wang et al. (2015) XRD data from HPbI₃ is also present and authors may comment on the similarities or differences.

Secondly, authors might discuss the work of Li et al. (2015) as it is connected to their work.

Reply: We greatly appreciate the referee's recommendation for carrying out single-crystal XRD to elucidate the crystal structure of HPbI₃. In the previous version of our manuscript, we thought that HI (HPbI₃) played a similar role to DMF/DMSO ($\text{PbI}_2 \cdot \text{DMF}$; $\text{PbI}_2 \cdot \text{DMSO}$) as a ligand in coordination to the Pb(II) center, and thus we believed that HPbI₃ has a similar structure to $\text{PbI}_2 \cdot \text{DMF}$ or $\text{PbI}_2 \cdot \text{DMSO}$. We have grown a crystalline sample of good quality by diffusion of chlorobenzene vapor into a DMF solution of 1.5:1 HI:PbI₂, and single crystal XRD

established its structural formula as $\text{HPbI}_3 \cdot \text{DMF}$, which belongs to hexagonal space group $P6_3mc$ (No. 186) with $Z = 2$. The crystal structure of $\text{HPbI}_3 \cdot \text{DMF}$ features a hexagonal array of anionic columns each composed of octahedral $[\text{PbI}_6]$ building units that are stacked along their opposite triangular faces, with highly-disordered DMF occupying the inter-columnar space. Unlike other solvates ($\text{MAPbI}_3 \cdot \text{DMF}$ and $\text{MAPbI}_3 \cdot \text{H}_2\text{O}$), $\text{HPbI}_3 \cdot \text{DMF}$ readily releases DMF at room temperature without any noticeable change of crystal structure due to the robustness of face-sharing unit and its hexagonally-packed coordination columns; the powder XRD pattern comparing exactly matched peaks is shown in revised Fig. 3c and Fig. S4 (or see below). TGA measurement revealed that the mass loss ratio of both powders and dry single crystals was about $\sim 28\%$, which was equal to 1:1 molar ratio of $\text{HI}:\text{PbI}_2$, thus conforming the molecular formula of HPbI_3 powders.

Figure S4. Crystal structure of $\text{HPbI}_3 \cdot \text{DMF}$ (highly disordered DMF solvate molecule and H^+ ions are not shown, the mobile H^+ ions are expected to be located around $[\text{PbI}_3]^-$ columns after DMF release). (a) Top view of hexagonal crystal structure and primitive cell (arrow indicates the (001) face-sharing unit); (b) side view of $\text{HPbI}_3 \cdot \text{DMF}$ super cell; (c) optical micrograph of as-prepared $\text{HPbI}_3 \cdot \text{DMF}$ crystal (width $\sim 2\text{mm}$); (d) simulated powder XRD patterns for HPbI_3 , $\text{HPbI}_3 \cdot \text{DMF}$ and the measured XRD pattern of HPbI_3 powders.

See the description below as given in page 3 of our revised manuscript:

“Single crystal X-ray analysis of $\text{HPbI}_3 \cdot \text{DMF}$ showed that its structure features a hexagonal array of $[\text{PbI}_3]^-$ anionic columns each composed of $[\text{PbI}_6]$ coordination octahedra that are stacked along their opposite triangular faces (Supplementary Fig. 4). The inter-columnar space is sufficiently large to be filled by H^+ ions and an equimolar equivalent of highly disordered DMF molecules. Unlike other solvated $[\text{PbI}_3]^-$ compounds (e.g. $\text{MAPbI}_3 \cdot \text{DMF}$, $\text{MAPbI}_3 \cdot \text{H}_2\text{O}$), the DMF component in $\text{HPbI}_3 \cdot \text{DMF}$ can be removed without changing its robust facet-stacking $[\text{PbI}_3]^-$ structure. After washing with diethyl ether or setting aside for several minutes, the crystal sample recorded a $\sim 27.8\%$ weight loss in TGA, which confirms the molecular formula of HPbI_3 . Presumably the protons in HPbI_3 can move freely in the intervening space between hexagonal arrays of anionic coordination columns. Powdery HPbI_3 as starting material for device fabrication was prepared following the literature procedure,³¹ with careful modification using ethanol

instead of diethyl ether to remove excess HI and precipitate our products, followed by air pump filtration and drying at 60 °C overnight for purification. It was then re-dissolved in DMF solution and different amounts of CH₃NH₂ ethanol solution were freshly added to obtain the NABR perovskite precursors. The resulting HPbI₃ sample washed by diethyl ether did not exhibit high stability according to our control experiment. ”

See the method below as given in Page 7.

Crystal Structure Analysis: X-Ray intensities of HPbI₃·DMF were collected at 296 K on a Bruker AXS Kappa Apex II Duo diffractometer with MoK_α radiation ($\lambda = 0.71073 \text{ \AA}$) from a sealed-tube generator. Crystal data: hexagonal, $a = b = 8.7339(10)$, $c = 8.1770(11) \text{ \AA}$, and $Z = 2$. The systematic absences are consistent with both noncentric space group $P6_3mc$ (No. 186) and centric space group $P6_3mcm$ (No. 193). As the proton has negligible X-ray scattering and the DMF molecule exhibits severe orientational disorder, structure determination was based on space group $P6_3mc$ with only the Pb and I atoms subjected to anisotropic least-squares refinement using the *SHELXL-97* program; 7944 reflections measured, of which 509 are unique, $R_{int} = 0.099$, $RI = 0.044$, $wR2 = 0.134$, and $GOF = 1.05$. The crystallographic data for this paper have been deposited with the Cambridge Crystallographic Data Centre (CCDC) as No. 1479488. These data can be obtained free of charge from CCDC via www.ccdc.cam.ac.uk/data_request/cif.

See the revised Figure 3.

Figure 3. Stages of crystalline phase conversion. **a**, The starting PbI_2 and HPbI_3 materials, as well as controlled perovskite film by one step method. **b**, NABR prepared perovskite films with increasing amounts of CH_3NH_2 : 0.3, 0.35 and 0.4 mL MA in 1.5 M 1mL HPbI_3 . **c**, XRD patterns of PbI_2 (bottom) and HPbI_3 (top) films at different stages of reaction: dip-coated wet film, spin-coated films before baking and after baking. **d**, XRD patterns of $\text{CH}_3\text{NH}_3\text{PbI}_3$ films by control (bottom) and NABR (top) methods at different reacting stages: dip-coated wet film, spin-coated films before baking and after baking. **e**, Crystallographic illustration of sequential NABR conversion from PbI_2 , to $\text{PbI}_2\cdot\text{DMF}$, to HPbI_3 (H^+ ions are included to indicate stoichiometry but are actually mobile around $[\text{PbI}_3]$ column), then to intermediate $\text{CH}_3\text{NH}_3\text{PbI}_3\cdot\text{DMF}$, and finally to $\text{CH}_3\text{NH}_3\text{PbI}_3$.

To avoid similar oversight and misinterpretation, we have carefully indexed the XRD patterns of $\text{MAPbI}_3\cdot\text{H}_2\text{O}$ and $\text{MAPbI}_3\cdot\text{DMF}$ to check the transition products as well (See Fig. 3 and Fig. S1 or below).

Figure S1. (a) Degradation and recovery mixed $\text{CH}_3\text{NH}_3\text{PbI}_{3-x}\text{Cl}_x$ perovskite using methylamine. (b) Crystal model of $\text{MAPbI}_3\cdot\text{H}_2\text{O}$ (c) Crystal model of $\text{MAPbI}_3\cdot\text{DMF}$ (d) Simulated XRD for verification of transition products using crystal models. Note: The mixed $\text{CH}_3\text{NH}_3\text{PbI}_{3-x}\text{Cl}_x$ perovskite films were sensitive to moisture and the recovery characterization was a little difficult because of some monohydrate in the recovered film.

We have also compared our finding with the previous work of Wang *et al.* and found that HPbI_3 in that report still contained some PbI_2 impurity. Our modified procedure produces stable HPbI_3 and thus avoids contamination by impurities.

Our synthetic route produces much more stable intrinsic perovskite than Li's work (Ref. 20), as discussed in our paper. See the following text in the revised manuscript: "Besides, it is notable that stable perovskite solar cells through layer shielding reported in previous works are generally lower than 16.2% efficiency.^{20,38} According to the detailed reports, perovskite itself could not endure high humidity (>55%) for 1-6 days without encapsulation of its high-efficiency photovoltaics, which indicates that its high stability primarily comes from encapsulation. However, here we demonstrate that NABR using excess CH_3NH_2 to react with well-defined HPbI_3 provides a reliable route to producing material-stable perovskites superior than those reported in previous work.^{20,38}"

During the perovskite formation, the general intermediate compound is $\text{MAPbI}_3\cdot\text{DMF}$ from both traditional

MAI+PbI₂ recipe and our NABR MA+HPbI₃ method using DMF as solvent. The vacancies by traditional methods will allow the penetration of moisture molecules and form H-bonding with MA in the form of monohydrate. We employed HPbI₃ as starting material, which had identical coordination numbers between Pb and I ([PbI₃]). Sequential production of HPbI₃ and MAPbI₃ using NABR ensures well-defined coordination and stoichiometric ratio, thus resulting in high material stability due to reduced vacancy. Point 2:

Also, some minor inconsistencies in the text are confounding. For example, the phrase "hydrated-HPbI₃ (CH₃NH₃PbI₃·H₂O)" doesn't make much sense. Moreover, in the Discussion paragraph, one can see the following: "...under heating stress, releasing fade-away CH₃NH₂ molecule, then PbI₂ and HI vapor". At which temperature were the degradation experiments conducted so to produce the PbI₂ vapor?

Reply: We have thoroughly revised our manuscript to remove errors in English usage and inconsistent text.

We have changed "hydrated-HPbI₃ (CH₃NH₃PbI₃·H₂O)" to "monohydrate".

We intended to say PbI₂ solid and HI vapor, thus we have changed "PbI₂" to "PbI₂ solid".

Point 3:

The paper is lucid in general terms, however, some minor and major queries, mentioned above, are aimed to improve the credibility of the paper.

To conclude, the reason for publishing the paper is to report about outstanding long-lasting performance in the presence of humidity and new interesting way of MAPbI₃ synthesis. This should probably be the keynote of the paper and compared more to other methods of protracting the working time of the MAPbI₃ thin-films. On the other hand, as it has been already mentioned in the first revision, the conclusions made on the structure of the intermediate compound, HPbI₃ are brusque and based only on XRD and UV-absorptions measurements, which suggest that the results can't be fully interpreted. The previous revision didn't improve the paper to the level when it should be published in the journal with such high credibility. On the other hand, one can't deny the results are curious and interesting, though they are in the nascent form, and should be definitely published if not in Nature Communications then elsewhere.

Reply: We thank the reviewer again for the comprehensive comments. We have accordingly determine the crystal structure of HPbI₃·DMF and also assigned the intermediate compounds based on XRD checking and fitting. **Besides, we have revised the paper thoroughly to enhance its presentation.** We hope our revised manuscript meets the high standard for publication in Nature Communications.

Reviewer #3 (Remarks to the Author)

I would like to thank authors for their endeavors to enlighten the insights of the proposed one-step mechanism. Specifically, in response to my previous query, authors have conducted detailed XRD experiments to determine the structure of intermediate compound HPbI₃*DMF. However, the initial query was about the XRD structure determination of HPbI₃ itself, if possible. As the result, the revised manuscript is more confusing in terms of crystal structure explanation.

The question of the correct understanding of the proposed HPbI₃ intermediate structure remains ambiguous. For example, the implication 'Unlike other solvated [PbI₃] compounds (e.g. MAPbI₃·DMF, MAPbI₃·H₂O), the DMF component in HPbI₃·DMF can be removed without changing its robust facet-stacking [PbI₃] structure' can not be substantiated due to the lack of experimental evidence. How can authors prove the full loss of DMF resulting in HPbI₃? Is it possible to detect it via TGA analysis? In the main text, it is asserted that a 28% weight loss confirms HPbI₃ structure, whereas it is not completely clear what this 28% exactly comprise of: HI, DMF, or both HI and DMF? Any theoretical calculations did not yield 28% loss in TGA. Could authors explain this with more details?

On the other hand, it is very likely that HPbI₃*DMF may adopt a hexagonal structure; however, authors should understand that the structure of HPbI₃*DMF does not necessarily allow for the same structure of HPbI₃, if the latter exists. Remarkably, in the previous versions of the manuscript, HPbI₃ was claimed to adopt a monoclinic structure: 'HPbI₃ has full coordination and adopts a monoclinic structure employing [PbI₃] double chain with edge sharing'. This illustrates how speculative the discussed matter is. In my opinion, unless HPbI₃ itself is confirmed through a reliable analysis, no statements concerning its structure or existence should be made at all. In addition, Figure S4 (d) depicts the XRD patterns of both simulated and experimental HPbI₃, whereas no experimental pattern for HPbI₃*DMF is presented. Moreover, a detailed crystal structure determination data for HPbI₃*DMF in Supplementary might increase the credibility of the analysis.

In general, my opinion is that the goal of the publication initially should have been a report about a novel method of MAPbI₃ thin-films formation, while an attempt to go into more details about the mechanism and structures of intermediate compounds implicated difficulties and obscured the initial importance of the performance of thin-films.

Response to Reviewers' comments:

Reviewer #3 (Remarks to the Author):

1. I would like to thank authors for their endeavors to enlighten the insights of the proposed one-step mechanism. Specifically, in response to my previous query, authors have conducted detailed XRD experiments to determine the structure of intermediate compound $\text{HPbI}_3 \cdot \text{DMF}$. However, the initial query was about the XRD structure determination of HPbI_3 itself, if possible. As the result, the revised manuscript is more confusing in terms of crystal structure explanation.

Answer: We feel sorry that we made it more confusing in terms of the HPbI_3 structure after we talked about the $\text{HPbI}_3 \cdot \text{DMF}$ (We call it “ HPbI_3 solvate” in the revised manuscript). In previous revision, to answer the previous query about the XRD structure determination of HPbI_3 itself, we dissolved 1:1.5 of PbI_2/HI in DMF solution, with anti-solvent CBZ (chlorobenzene) vapor diffusion into the system to prepare the single crystals, and got the single crystal of semitransparent HPbI_3 solvate first. When we heated the semi-transparent crystal (HPbI_3 solvate) at 80 °C to remove the solvent, we found it became faint yellow quickly, which indicated it might convert to HPbI_3 single crystal. Actually, we characterized faint yellow crystal before HPbI_3 solvate using single crystal XRD and confirmed it indeed had hexagonal structure, 1/3 Pb/I atomic ratio, as well as ~28% ratio of HI/ PbI_2 revealed by TGA. During analyzing the crystallographic structure of HPbI_3 , we found there was too large space between the hexagonally arranged $[\text{PbI}_3]^-$ columns (density: HPbI_3 , 2.2 g/cm³; MAPbI_3 , 4.2 g/cm³; PbI_2 : 6.2 g/cm³). We continued to characterize the single crystal XRD for semi-transparent HPbI_3 solvate directly from the mother liquor and tried to gain some important formation insight into HPbI_3 . The characterization suggested they had nearly the same arrangement of $[\text{PbI}_3]^-$ columns.

We thought HPbI_3 was converted from HPbI_3 solvate and thus discussed the HPbI_3 solvate before HPbI_3 in the previous version, without statement that the following discussion on HPbI_3 was also based single crystal XRD.

The DMF solvent facilitated the lattice formation and after releasing solvent the same $[\text{PbI}_3]^-$ structure was kept. Right now we emphasize the structure of HPbI_3 based on the referee's query and revise the manuscript for clear demonstration. The HPbI_3 solvate is just discussed in Fig. S4 and METHODS. See the red in page 3 or below.

“Single crystal X-ray analysis of faint yellow HPbI_3 crystal shows a hexagonal array of $[\text{PbI}_3]^-$ anionic columns each composing of $[\text{PbI}_6]$ coordination octahedra that are stacked along their opposite triangular faces, with exact molecular formula HPbI_3 (see METHODS and Supplementary Fig. 4).”

Figure S4. Crystal information of HPbI₃ and HPbI₃ solvate based on single crystal XRD. (a) Top view of hexagonal [PbI₃]⁻ columns and its primitive cell (arrow indicates the (001) face-sharing unit, which seems to be mechanically robust in the form of tripod-stacking) and (b) side view of super cell (highly disordered solvent molecule and H⁺ ions are not shown, the mobile H⁺ ions are expected to be located around [PbI₃]⁻ columns after solvent release); (c) Top view of hexagonal HPbI₃ solvate. In order to show the disorder of DMF in the lattice, the occupation of 3 -N-(CH₃)₂ groups in DMF is set to 1/3, O of DMF with proton nearby is set to 1/2 occupation on both sides of close center C, and close center C atom (it is amplified for observation) of DMF (HO-C-N(CH₃)₂) is 100% occupation for simplicity; (d) optical micrograph of as-prepared HPbI₃ solvate crystal (width ~2mm) (After heating at 80 °C for 30 minutes, it loses DMF and becomes HPbI₃); (e) Simulated powder XRD patterns for HPbI₃, HPbI₃ solvate based on single crystal XRD produced crystal information, as well as the experimental XRD pattern for HPbI₃ powder (spin-coated) and HPbI₃ solvate powder (grounded from one ~30 mg crystal).

2. The question of the correct understanding of the proposed HPbI₃ intermediate structure remains ambiguous. For example, the implication 'Unlike other solvated [PbI₃] compounds (e.g. MAPbI₃·DMF, MAPbI₃·H₂O), the DMF component in HPbI₃·DMF can be removed without changing its robust facet-stacking [PbI₃] structure' can not be substantiated due to the lack of experimental evidence.

Answer: Based on the single crystal XRD, the proposed HPbI₃ is clear judging from 1:3 Pb/I ratio and the same structure as HPbI₃ solvate.

Both HPbI₃ solvate and HPbI₃ have the hexagonal [PbI₃]⁻ columns, each of which composes of [PbI₆] coordination octahedra that are stacked along their opposite triangular facets. This tripod-stacking makes the column robust enough after releasing the solvents. Therefore, "...the DMF component in HPbI₃·DMF

can be removed without changing its robust facet-stacking [PbI₃] structure” is actually supported by single crystal XRD. In this version, the discussion about this is deleted because we focus on the HPbI₃.

3. How can authors prove the full loss of DMF resulting in HPbI₃? Is it possible to detect it via TGA analysis? In the main text, it is asserted that a 28% weight loss confirms HPbI₃ structure, whereas it is not completely clear what this 28% exactly comprise of: HI, DMF, or both HI and DMF? Any theoretical calculations did not yield 28% loss in TGA. Could authors explain this with more details?

Answer: First, the single crystal XRD gives the 1/3 Pb/I atomic ratio for both HPbI₃ and HPbI₃ solvate. Second, the powdery HPbI₃ in this work was washed by ethanol and stored in 80°C for over 6 months. Our TGA result for the powder suggests weight ratio of HI:PbI₂ is 27.8% judging from the weight loss value at ~350 °C (Fig. S13), which is consistent with ~28% HI:PbI₂ weight ratio in one single crystal after DMF loss (Fig. S24). Due to our mispresentation (“28% weight loss” actually meant to express “28% weight ratio of HI/PbI₂ judging from the weight loss”), we feel sorry that it caused a misunderstanding of the right TGA data in previous version. We hope the present version has improved the explanation. Thank again for the referee’s comments.

Figure S13. Thermal behavior and stability. (a) TGA curves of CH₃NH₃I, PbI₂ and perovskites prepared by one-step, two-step and NABR methods. (b) TGA curves of four different perovskites with 30 min heat preservation at 100 °C and 30 min heat preservation at 100 °C.

120 min heat preservation at 200 °C. Note: weight loss was normalized to [0, 1]. 1:1 MAI:PbI₂ control sample have small weight loss at the onset (lower than 50 °C).

4. On the other hand, it is very likely that HPbI₃*DMF may adopt a hexagonal structure; however, authors should understand that the structure of HPbI₃*DMF does not necessarily allow for the same structure of HPbI₃, if the latter exists. Remarkably, in the previous versions of the manuscript, HPbI₃ was claimed to adopt a monoclinic structure: 'HPbI₃ has full coordination and adopts a monoclinic structure employing [PbI₃] double chain with edge sharing'. This illustrates how speculative the discussed matter is. In my opinion, unless HPbI₃ itself is confirmed through a reliable analysis, no statements concerning its structure or existence should be made at all.

In addition, Figure S4 (d) depicts the XRD patterns of both simulated and experimental HPbI₃, whereas no experimental pattern for HPbI₃*DMF is presented. Moreover, a detailed crystal structure determination data for HPbI₃*DMF in Supplementary might increase the credibility of the analysis.

Answer: We thank the referee for agreeing that HPbI₃ solvate adopts a hexagonal structure. As above mentioned, the conclusion on hexagonal HPbI₃ is also based on single crystal XRD, although we don't emphasize it in previous version. The fitted powder XRD patterns for both come from the single crystal XRD. Our single crystal XRD has revealed they have the same [PbI₃]⁻ columns in the lattice for the two.

Besides, in response to the comment, we have added the experimental powder XRD pattern for HPbI₃ solvate, which was grounded from one large crystal (~30 mg). Please see below, we thank the referee for the careful review.

The detailed crystal information files (.cif) for HPbI₃ and HPbI₃ solvate from single crystal XRD have been enclosed in Supplementary. We also update the Fig S4 to give more details for illustrating the crystal information for HPbI₃ solvate. In order to show its disorder of DMF in the lattice, the occupation of 3 -N-(CH₃)₂ groups in DMF is set to 1/3, O of DMF with proton nearby is set to 1/2 occupation on both sides of close center C, and close center C atom (it is amplified for observation) of DMF (HO-C-N(CH₃)₂) is 100% occupation for simplicity. This XRD simulation has the same pattern as HPbI₃ because the disorder part does not contribute XRD.

Figure S4. Crystal information of HPbI₃ and HPbI₃ solvate based on single crystal XRD. (a) Top view of hexagonal [PbI₃] columns and its primitive cell (arrow indicates the (001) face-sharing unit, which seems to be mechanically robust in the form of tripod-stacking) and (b) side view of super cell (highly disordered solvent molecule and H⁺ ions are not shown, the mobile H⁺ ions are expected to be located around [PbI₃] columns after solvent release); (c) Top view of hexagonal HPbI₃ solvate. In order to show the disorder of DMF in the lattice, the occupation of 3 -N-(CH₃)₂ groups in DMF is set to 1/3, O of DMF with proton nearby is set to 1/2 occupation on both sides of close center C, and close center C atom (it is amplified for observation) of DMF (HO-C-N(CH₃)₂) is 100% occupation for simplicity; (d) optical micrograph of as-prepared HPbI₃ solvate crystal (width ~2mm) (After heating at 80 °C for 30 minutes, it loses DMF and becomes HPbI₃); (e) Simulated powder XRD patterns for HPbI₃, HPbI₃ solvate based on single crystal XRD produced crystal information, as well as the experimental XRD pattern for HPbI₃ powder (spin-coated) and HPbI₃ solvate powder (grounded from one ~30 mg crystal).

5. In general, my opinion is that the goal of the publication initially should have been a report about a novel method of MAPbI₃ thin-films formation, while an attempt to go into more details about the mechanism and structures of intermediate compounds implicated difficulties and obscured the initial importance of the performance of thin-films.

Answer: Thanks for the positive comments on the novel method. Right now, we have also addressed

the referee's concern about mechanism and structures of intermediate compounds.

In summary, we develop NABR route and address the proposed mechanism through intermediate compound structure characterization. The insight into synthesis route should shed light on the further research for the improvement of intrinsic stability of perovskite materials. As you know, the stability of MAPbI₃ is more challenging than other perovskites (such as inorganic perovskite or their mixed counterpart) and our NABR have addressed this problem to great extent. We hope it can be qualified to publish in Nature Communications in present version with appreciation.

Reviewer #1 (Remarks to the Author)

The authors have responded to all referee comments adequately and included new single-crystal XRD data on the structure of HPbI₃, which the perovskite community will find very useful. I recommend the publication of this manuscript without delay.